



# The Elbrus (Caucasus, Russia) ice core glaciochemistry to reconstruct anthropogenic emissions in central Europe: The case of sulfate

Susanne Preunkert[1,2], Michel Legrand[1,2], Stanislav Kutuzov[3], Patrick Ginot[1,2,4], Vladimir Mikhalenko[3], and Ronny Friedrich[5]

[1]Université Grenoble Alpes, CNRS, Institut des Géosciences de l'Environnement (IGE), Grenoble, 38402, France

[2]CNRS, Institut des Géosciences de l'Environnement (IGE), Grenoble, 38402, France

[3]Institute of Geography, Russian Academy of Sciences, Moscow, 119017, Russia

[4]Observatoire des Sciences de l'Univers de Grenoble, IRD/UGA/CNRS, Grenoble, 38400, France

[5]Curt-Engelhorn-Center Archaeometry, Mannheim, Germany

*Correspondence to*: Susanne Preunkert (suzanne.preunkert@univ-grenoble-alpes.fr)

**Abstract.** This study reports on the glaciochemistry of a deep ice core (182 m long) drilled in 2009 at Mount Elbrus (43°21′N, 42°26′E; 5115 m above sea level) in the Caucasus, Russia. Radiocarbon dating of the particulate organic carbon fraction in the ice suggests a basal ice age of ~1670 ± 400 cal yr BP. Based on chemical stratigraphy, the upper 168.6 m of the core were dated by counting annual layers. The seasonally resolved chemical records cover the years 1774-2009 (Common Era), thus, being useful to reconstruct many aspects of atmospheric pollution in central Europe from pre-industrial times to present-day. After having examined the extent to which the arrival of large dust plumes originating from Sahara and Middle East modifies the chemical composition of the Elbrus (ELB) snow and ice layers, we focus on the sulfur pollution. The ELB sulfate levels indicate a four- and six-fold increase from 1774-1900 to 1980-1995 in winter and summer, respectively. Remaining close to 116 ± 28 ppb during the nineteen century, the summer sulfate levels started to rise at a mean rate of ~6 ppb per year from 1920 to 1950. The summer sulfate increase accelerated between 1950 and 1975 (11 ppb per year), levels reaching a maximum between 1980 and 1990 (730 ± 152 ppb) and subsequently decreasing to 630 ± 130 ppb at the beginning of the twenty first century. Long-term sulfate trends observed in the ELB ice cores are compared with those previously obtained in Alpine ice, the most important difference consists in a more pronounced decrease of the sulfur pollution over the three last decades in western than central Europe.



## 1 Introduction

It is now well recognized that the present climate change is not only related to change of long-lived greenhouses gases but also to aerosols, particularly at regional scales. In this way, it has been suggested that aerosols may have weakened the rate of the global warming during the second part of the last century (Andreae et al., 2005). However, uncertainties still exist in
quantifying the climatic impact of aerosols, because the spatial distribution of aerosols is very heterogeneous and requires therefore numerous observations to make these parameters useful as inputs and constraints for climate models. An important gap is also related to the fact that direct atmospheric observations are available only from far later (starting with the appearance of the acid rain phenomena in the late 1960s) than man-made activities started to disturb the pre-industrial atmosphere. However, to predict future climate the knowledge of atmospheric changes in aerosol load and composition from
present-day polluted atmosphere back to preindustrial times is required. Chemical records of species trapped in snow deposited on cold glaciers provide a unique and powerful way to reconstruct past atmospheric chemistry changes including aerosol load and composition (see Legrand and Mayewski (1997) for a review).

In Europe, a largely industrialized continent, ice cores were extracted from high-elevation glaciers located at various places, including the Alps (Preunkert and Legrand, 2013; Schwikowski et al., 2004), the continental Siberian Altai (Eichler et al.,
2011; Olivier et al., 2006), and Kamchatka (Kawamura et al., 2012). In the Alps, intimately connected to western European emissions, ice cores have been performed at Col du Dôme (CDD, Mont Blanc, Preunkert, 2001), Fiescherhorn (Bernese Alps, Jenk, 2006), and Colle Gnifetti (CG) in the Monte Rosa region (Schwikowski, 2006; Wagenbach et al., 2012) in view to examine various aspects of atmospheric pollution. The exceptionally high net snow accumulation at the CDD site permitted the extraction of seasonally resolved records of various chemical species over the last 100 years (Preunkert and
Legrand, 2013). In older ice layers preservation of winter layers at CDD becomes very limited and summer layers become very thin. Conversely, the low and incomplete net snow accumulation rate at CG, which is controlled by wind erosion, highly limits the preservation of winter ice layers (Wagenbach et al., 2012), but is low enough to provide access to an extended time period, at least over the last millennium. Using the EMEP (European Monitoring and Evaluation Programme) regional chemistry-transport model and past emission inventories of SO2 in Europe, observed CDD long-term trends of
sulfate were fairly well reproduced, leading Fagerli et al., (2007) to conclude that the seasonal changes seen at the CDD alpine site are associated with geographical changes in source regions impacting the site. This is a strong argument for a separate examination of summer and winter data, extracted from alpine ice cores. However, until now, Alpine ice cores document only the last hundred years (at the best back to 1890, Legrand et al., 2018) on a seasonal basis, whereas the early stage of the industrialization time period, which is generally considered to have started around 1850, is missed. An ice core
recently extracted from the Elbrus (the highest summit of the Caucasus) indicated excellent preservation of summer and winter layers at least back to 1820 (Mikhalenko et al., 2015). Thus, the ELB ice may contain very valuable information on past atmospheric pollution in central Europe since the beginning of the industrialization.




Here we report on the glaciochemistry of a deep ice core (182 m long) drilled in 2009 at Mount Elbrus in the Caucasus, Russia. The seasonally resolved chemical records were obtained back to 1774 (i.e., well prior to the onset of the industrial period). Data are discussed in two companion papers of which this one. The present paper examines first of all the impact of dust plumes, which arrive sporadically from Sahara and Middle East, on the chemical composition of the Elbrus (ELB) snow

and ice layers. It then focuses on long-term sulfate trends in relation to growing sulfur pollution. The long-term summer and winter trends of sulfate are discussed with respect to past $SO_2$ emissions in central Europe and compared to those extracted at the Alpine site of CDD in relation to $SO_2$ emissions from western Europe. The second paper focuses on calcium (a dust tracer) long-term trend (Kutuzov et al., this issue), discussing its past changes in relation with natural variability, as well as climatic and land use changes in the dust source regions Middle East and North Africa.

## 2 Methods and Dating

### 2.1 Discrete Subsampling of firn and ice, and Chemical Analysis

Pieces of cores were cleaned under a clean air bench located in a cold room (-15°C) using an electric plane tool previously developed to process Alpine firn and ice samples. A total of 3724 subsamples were obtained along the upper 168.6 m of the Elbrus core. The depth resolution decreased from 10 cm at the top to 5 cm at 70 m, and 2 cm at 157 m depth and below.

Given the decrease of the net annual snow accumulation from 1.5 mwe (0.8 mwe in summer and 0.7 mwe in winter) near the surface to 0.18 mwe (0.15 mwe in summer and 0.03 mwe in winter) at 157 m depth, as seen in Figure 1, this sampling permitted to minimize the lost of temporal resolution with depth along the core, particularly in summer. In this way, an average of 9 summer samples per year were sampled at 157 m depth (compared to 15 summer samples per year near the surface). The larger decrease of the net snow accumulation in winter than in summer leads to a more pronounced loss of

resolution in winter layers (12 samples per winter near the surface and 1-2 samples per winter at 157 m depth)(Figure 1).
For measurements of cations ($Na^+$, $K^+$, $Mg^{2+}$, $Ca^{2+}$, and $NH_4^+$), a Dionex ICS 1000 chromatograph equipped with a CS12 separator column was used. For anions, a Dionex 600 equipped with an AS11 separator column was run with a quaternary gradient of eluents ($H_2O$, NaOH at 2.5 and 100 mM, and $CH_3OH$). A gradient pump system allows the determination of inorganic species ($Cl^-$, $NO_3^-$, and $SO_4^{2-}$) as well as short-chain carboxylates. Investigated carboxylates include formate

($HCO_2^-$), lactate ($CH_3CHOHCO_2^-$), acetate ($CH_3CO_2^-$), glycolate ($CH_2OHCO_2^-$), and glyoxylate ($CHOCO_2^-$), oxalate ($C_2O_4^{2-}$), malate ($CO_2CH_2CHOHCO_2^{2-}$), malonate ($CO_2CH_2CO_2^{2-}$), succinate ($CO_2(CH_2)_2CO_2^{2-}$), and glutarate ($CO_2(CH_2)_3CO_2^{2-}$). Details on working conditions are reported in Legrand et al. (2013). For all investigated ions, blanks of the ice decontamination procedure were found to be insignificant with respect to respective levels found in the ice cores.
During the drill operations, an incident occurred at the depth of 31 m and a fluid was poured in the hole to liberate the drill

device. This has led contamination of the firn at 31 m down to the firn-ice transition located at 55.7 m depth. Samples covering the 1983-1997 years were contaminated for sodium (124 ± 87 ppb compared to 26 ± 28 ppb over the 16 preceding years) and potassium (35 ± 25 ppb compared to 16 ± 15 ppb over the 1966-1982 years). One core section (denoted ELB-140)





that covers winter 1875/76, summer 1876, and winter 1876/1877 was not analysed. Finally, a part of the ELB-138 ice core section that covers winter 1877/1878 was of poor quality (splitted ice).

## 2.2 Annual Layer Counting

As discussed by Mikhalenko et al. (2015), dating of the Elbrus ice can be done by annual layer counting on the basis of the
stratigraphic ammonium and succinic acid records, both exhibiting well-marked winter minima. As previously seen in alpine ice cores, ammonium reveals a well-marked maximum in summer due to a maximum of $NH_3$ emission together with an efficient upward transport in summer (Fagerli et al., 2007). Succinic acid is a light dicarboxylic acid for which a strong summer maximum and a quasi-null winter level can be observed in the present-day atmosphere in Europe (Legrand et al., 2007). The very low winter levels of this organic compound are related to the absence of a winter source of this species,
which is mainly photo-chemically produced from biogenic precursors. Mikhalenko et al. (2015) assumed a concentration limit of 100 ppb ammonium and 5 ppb of succinate to separate winter and summer in the upper layers down to 75.6 m depth (i.e., 1963). To account for an observed decreasing trend of ammonium concentrations with depth (i.e., due to a post 1950 increase as also seen for species like nitrate and sulfate, see below), the ammonium winter criterion was adjusted to 50 ppb between 75.6 and 86.8 m depth (i.e., 1950-1963) and 30 ppb below. Since no systematic change of succinate with depth is
observed, the succinate concentration limit of 5 ppb was also applied in deeper layers. In this way, the annual counting was found to be very accurate dating (a 1-year uncertainty) over the last hundred years when anchored with the stratigraphy with the Katmai 1912 horizon (Mikhalenko et al., 2015). Though the annual counting becomes less evident prior to 1860, Mikhalenko et al. (2015) reported an ice age of 1825 at 156.6 m depth, what is still consistent with the presence of volcanic horizon at around 1833-1940 such as Coseguina (1835).
We here extended the annual counting down to 168.5 m depth (i.e., 131.6 mwe) by considering a concentration limit of 30 ppb for ammonium and 5 ppb for succinate. With that, ice dates to 1774 CE at the depth of 168.5 m. In the following we will examine individual half-year summer and winter values as well as monthly means. To calculate monthly means, a uniform snowfall rate is assumed within each half-year. Winter samples were attributed to the last 3 months of the year and the three first months of the following year (i.e., winter 1850/1851 is from 1850.75 to 1851.25). Summer samples are from the fourth
to the ninth month (i.e., summer 1850 is from 1850.25 to 1850.75) in each year. In Figure 2 we report the obtained chronology for three different sequences including the deepest one (1774-1784). It could be seen that in the years prior to 1850, quite often a winter layer is made only of one or two samples, whereas summer layers are made still of more than 6 samples. Below 168.5 m depth, the ice core quality becomes rather bad (numerous small pieces of broken ice) rendering subsampling and ice decontamination not evident. Furthermore, as seen in Figure 3, in contrary to what is observed above
168,5 m, the 30 subsamples obtained along a 1 m long core section at 176.3-177.3 m depth reveal an absence of samples with ammonium and succinate concentrations below the applied winter criterion. This hampered the dating of the basal ice layers of the core by annual counting and therefore another dating approach based on [14]C of particulate organic matter was made for this part of the core.



### 2.3 Basal Ice Dating

Four ice samples located along the deepest 6 m of the core (bottom at 182.65 m, 142.1 mwe) were analysed for radiocarbon in particulate organic carbon (PO$^{14}$C). The lowest 0.5 m of the core were not analyzed due to a large presence of macro-size inorganic particles. To minimize the time interval covered by each sample, sample lengths were kept as short as possible

with respect to the detection limit related to working conditions during sampling and analysis. A typical sample length of 26 to 40 cm (available ice core section of 14-21 cm$^2$) was used leading to an initial ice sample mass of 430-580 g. After having decontaminated ice sample (using a DOC decontamination method according to Preunkert et al., 2011), melted ice was filtered and combusted at 340°C at the Institut de Géophysique Externe (Grenoble) using the inline filtration-oxidation-unit REFILOX (Reinigungs-Filtrations-Oxidationssystem, Hoffmann et al., 2018). Hereby, the resulting ice mass was reduced to

260-320 g containing 4.4 to 6.5 µgC of POC (Table 1). After cryogenic extraction of the $CO_2$ content, radiocarbon analyses were done at the accelerator mass spectrometer facility at the Curt-Engelhorn-Center Archaeometry (CEZA) in Mannheim (Hoffmann et al., 2017). Calibration of the retrieved 14C ages was done using OxCal version 4.3 (Bronk Ramsey, 1995).

To test the reliability of the DOC decontamination method (Preunkert et al., 2011) for POC analysis, a 340 °C REFILOX mass combustion comparison was made between ultrapure water and decontaminated blank ice (3 samples of 200 to 500

mL). To achieve an impurity-free solid ice the ultra-pure water was slowly frozen in polyethylene (PE) foil (Hoffmann et al., 2017 and references in there). The comparison showed that whatever the blank ice volume, the blank values were in the same range than the ultrapure water POC blanks (~0.4 ± 0.25 µgC) which were determined during the course of the ELB radiocarbon sample measurements. Thus, the ice decontamination procedure used for DOC ice measurement is also valid for POC ice decontamination.

Since the $CO_2$ collection line was recently extended to allow sample pooling, we were now able to directly determine the fraction modern carbon (F$^{14}$C) in blanks done with ultrapure water. A F$^{14}$C value of 0.71 ± 0.07 was found, measured on three blank samples in total (each consisting of four pooled samples). This value is in agreement with F$^{14}$C blank values found in previous studies (as reviewed and adopted in Hoffmann et al., 2018). In Table 1, we report ice sample data after blank correction, including correction of the F$^{14}$C value as well as correction of the extracted POC mass with the respective

ultrapure water blank determined before each ice sample extraction.

Using a mean mass related combustion efficiency of the device of 0.7 (Hoffmann et al., 2018), the mean POC concentrations of the four samples obtained by combustion at 340°C is of 25.4 ± 3.1 ngC g$^{-1}$ with highest concentration of 29.0 ngC g$^{-1}$ for the lowest sample analysed at 142.6 mwe (182.0 m). These values are in good agreement with those observed by Hoffmann et al. (2017) in the 340°C POC fraction of the lowest 8 mwe of a CG ice core (37 ± 16 ngC g$^{-1}$). Since a similarity between

CG and ELB was also observed for their preindustrial black carbon content (Lim et al., 2017), we thus exclude significant age errors due to a POC contamination during $^{14}$C sample preparation and analysis.

As seen in Table 1, the mean age of the ELB-178-03 sample (1530 yr cal BP) is older than the mean age of sample ELB-181-01 located 2.3 m below the ELB-178-03 sample. However, given age uncertainties, it is difficult to conclude that, as



observed previously at other mid-latitude glacier sites such as CG (Hoffmann et al., 2018), the ELB radiocarbon ages do not increase monotonically with depth as would be expected from well-behaved ice flow. PO$^{14}$C measurements suggest that the ELB ice core extends to ~ 1670 ± 400 yr cal BP (Table 1). This is younger than basal ice ages found at Alpine sites, i.e. ~5000 yr BP for Col du Dome (Preunkert et al., in press), ~ 4000 yr BP (Hoffmann et al., 2018) and >10,000 yr BP (Jenk et

al., 2009) for two CG ice cores, and ~7000 yr BP for Mt. Ortles (3905 m asl) (Gabrielli et al., 2016). From the observed temperature gradient in the borehole ELB site, Mikhalenko et al. (2015) calculated a heat flux at the bottom glacier that is 4-5 times larger than the mean value for the Earth's surface, possibly due to a heat magma chamber of the Elbrus volcano. That may lead to basal ice melting and removal of the oldest basal layers. If so, that may explain the young age of basal ice at the volcanic crater site compared to other non-volcanic mountain glaciers. The age of the basal ELB ice is nevertheless largely

greater than expected by ice flow model calculations, estimating a basal ice age of less than 400 years at the drill site (Mikhalenko et al. 2015).

### 3 The Effect of large Dust events on the Chemistry of ELB Ice

Large dust plumes originating from Middle East and less frequently from Sahara reach the Caucasus (Kutuzov et al., 2013). As seen in the Alps, these dust events disturb the chemistry of snow deposits, in particular with calcium rich alkaline snow

layers (Wagenbach et al., 1996). Depositions of these plumes disturb the level of numerous chemical species in Alpine ice because either they are present in dust at the emission stage or, being acidic, they were uptake by the alkaline dust material during transport (Usher et al., 2003). Preunkert (2001) showed that the arrival of dust plumes at CDD enhanced depositions of several cations (sodium, potassium, magnesium, and sodium) as well as acidic anions (sulfate, nitrate, chloride, fluoride, and carboxylates). To identify these layers in the ELB snow and ice we have estimated the acidity (or alkalinity) of samples

by checking the ionic balance between anions and cations with concentrations expressed in micro-equivalents per liter, µEq L$^{-1}$):

$$[H^+] = ([F^-] + [Cl^-] + [NO_3^-] + [SO_4^{2-}] + [MonoAc^-] + [DiAc^{2-}]) - ([Na^+] + [K^+] + [Mg^{2+}] + [Ca^{2+}] + [NH_4^+]) \qquad (1)$$

$$\text{with } [MonoAc^-] = [HCO_2^-] + [CH_3CHOHCO_2^-] + [CH_3CO_2^-] + [CH_2OHCO_2^-] + [CHOCO_2^-] \qquad (2)$$

$$\text{and } [DiAc^{2-}] = [C_2O_4^{2-}] + [CO_2CH_2CHOHCO_2^{2-}] + [CO_2CH_2CO_2^{2-}] + [CO_2(CH_2)_2CO_2^{2-}] + [CO_2(CH_2)_3CO_2^{2-}] \qquad (3)$$

In this work, samples which contain more than 120 ppb of calcium and which are below the 25% quartile of a robust spline through the calculated raw acidity profile, were considered as impacted by dust events. In this way, 616 (on a total of 2524) summer and 67 (on a total of 1150) winter samples were considered. Note that the results are quite similar when changing the calcium concentration criteria from 120 ppb to 100 or 140 ppb. Since the frequency of these events has changed over

time (Kutuzov et al., this issue) the significance of their impact on the deposition of chemical species is examined. Figure 4 shows the mean ionic budget of samples considered as contaminated by dust and containing high and moderate content of calcium compared to samples assumed to be free of dust events, over the 1950-1980 period. For the largest events (Ca$^{2+}$ >



600 ppb, Fig. 4A), the increase of calcium, accompanied by a strong increase of the alkalinity, reaches a factor of 7.4 compared to dust-free samples (Fig. 4C). In addition, this calcium enhancement is accompanied by an increase of a factor close to 8 for chloride, sodium, potassium and magnesium, whereas ammonium, nitrate, sulfate, and carboxylates are, at the best, enhanced by a factor of 2. When comparing dust samples containing weaker calcium contents (i.e. $Ca^{2+}$ < 600 ppb Fig.

4B) with dust-free samples (Fig. 4C), in addition to the increase of calcium (factor of 2) the most significant changes are seen for magnesium (x1.5), sodium and chloride (x1.6), and potassium (x1.3), respectively. In brief, all cations (except ammonium) and chloride are present in dust at the emission stage. Furthermore, acidic chloride can be taken up by dust during transport.

For species present in dust at the emissions stage, it is interesting to compare their ratio to calcium in ELB dust layers

(Figure 5) with atmospheric aerosol data obtained at sites impacted by dust events. For magnesium, a species predominantly originating from dust in ELB ice, the mean $[Mg^{2+}]/[Ca^{2+}]$ ratio in ELB dust events is 0.035 (Figure 5). Koçak et al. (2012) reported dust event related aerosol concentrations of sodium, magnesium, and calcium from two Eastern Mediterranean sites, i.e. from Erdemli (Turkey) with dust arriving from Middle East and from Heraklion (Crete) with dust from Sahara. It is important to emphasize that, as for the ELB ice data, the atmospheric concentrations of these cations correspond to their

water-soluble fraction (not the total fraction), which were measured with IC. In the case of Erdemli during Middle East dust events, Koçak et al. (2012) reported atmospheric concentrations of 7085 ng m$^{-3}$ for $Ca^{2+}$ and 423 ng m$^{-3}$ for $Mg^{2+}$ (Table 2). Since Erdemli is located at 22 m above sea level and 10 m away from the sea, in addition to the leachable fraction of magnesium from dust, a fraction of magnesium would here come from sea-salt. To correct concentration from the sea-salt contribution, we have used the $Na^+$ concentration (1148 ng m$^{-3}$) and assumed a mean $[Na^+]/[Ca^{2+}]$ ratio in dust of 0.08 as

observed in ELB ice samples containing dust (see below). Thus, neglecting the sea-salt calcium contribution, we estimate a dust sodium contribution of 567 ng m$^{-3}$ (0.08 x 7085 ng m$^{-3}$). With that, and using the $[Mg^{2+}]/[Na^+]$ ratio in seawater (0.12), we estimate that 70 ng m$^{-3}$ of magnesium are originated from sea-salt and calculate a $[Mg^{2+}]/[Ca^{2+}]$ ratio for dust aerosol close to 0.05 (Table 2). A similar value is obtained for aerosol at Heraklion during a Saharan dust event (0.043, Table 2). The content of $Mg^{2+}$ in ELB samples impacted by dust is therefore very consistent with what is observed in atmospheric

aerosol from the Eastern Mediterranean region during dust events. Note that the same is true for the $Mg^{2+}$ content of CDD samples impacted by Saharan dust (mean $[Mg^{2+}]/[Ca^{2+}]$ ratio of 0.045, Figure 5).

Figures 5 compares the chemical content of dust deposited in the Caucasus (ELB) and in the Alps (CDD). In the Alps, most of dust events are related to sporadic arrivals of Saharan dust plumes (Wagenbach et al., 1996). As discussed above, whereas Saharan dust events also sporadically reach the Elbrus site and are characterized by very large amount of calcium (see Fig.

4A) more frequent are dust events from Middle East that contain less calcium (Fig. 4B). As seen in Figure 5, ELB samples containing dust indicate a rather similar mean $[K^+]/[Ca^{2+}]$ ratio of 0.030 compared to dust events deposited in the Alps ($[K^+]/[Ca^{2+}]$ ratio of 0.035). The same remains true (not shown) for sodium ($[Na^+]/[Ca^{2+}]$ ratio of 0.08-0.09 at ELB and CDD sites) and chloride ($[Cl^-]/[Ca^{2+}]$ ratio of 0.11 for both ELB and CDD sites), suggesting that dust deposition of dust at the two sites is similar for these species present together with calcium at the emission stage. This conclusion is consistent with the





study of dust samples from several Middle East sites and Saharan showing similar chemical and mineralogical constituents in most cases (Engelbrecht et al., 2009).

For nitrate, and to a lesser extent for sulfate, a systematic lower content relative to calcium can be observed at the ELB site ($[NO_3^-]/[Ca^{2+}]$ ratio of 0.28 and $[SO_4^{2+}]/[Ca^{2+}]$ ratio of 0.72) compared to the French Alps site ($[NO_3^-]/[Ca^{2+}]$ ratio of 0.40

and $[SO_4^{2-}]/[Ca^{2+}]$ ratio of 0.90). Such a lower neutralisation of alkaline material by acidic species in dust plumes reaching the ELB site compared to the CDD site is probably related to a reduced availability of acidic species along the transport of the dust plumes towards the site

To evaluate the effect of dust on the deposition of chemical species, we compare in Table 5 averaged chemical concentrations of all samples with those not impacted by dust. Averages were obtained on the base of half-year summers

over the half-decades 1996-2000 and 1974-1978 characterized by high and low dust content, respectively (Kutuzov et al., this issue). Though the main impact of dust is as expected, on cations (except ammonium) and chloride, i.e. the constituents of dust particles, the impact is also significant for acidic species like nitrate and sulfate. For instance the increase of nitrate from 1974-1978 to 1996-2000 AD (144 ppb) in the whole dataset is largely (three quarters) related to the increase of dust as indicated by the smaller increase after removal of dust samples (increase of 31 ppb for the $NO_3^-_{red.}$ value). Same is true for

sulfate, for which the apparent increase between the two periods (90 ppb) is not due to an increase of pollution but of dust, as indicated by the drop of values when the dust free data set is considered (decrease of 48 ppb for the $SO_4^{2-}_{red.}$ value). The effect of changing dust inputs over time has to be therefore considered when discussing long-term trends in view to relate them to growing anthropogenic emissions (see Sect. 5 for sulfate in relation with $SO_2$ emissions). Finally, the large effect of dust seen for formate ($HCOO^-$) and not for acetate ($CH_3COO^-$) is in agreement with previous observations made by Legrand

et al. (2003) in Alpine ice and by (Legrand and De Angelis, 1995) in Greenland ice. These studies showed that the presence of formate and acetate in ice follows the uptake of formic and acetic acid from the atmospheric gas phase, and that the incorporation of these weak acids into hydrometeors is pH dependent with a stronger dependence for formic acid, which is a stronger acid than acetic acid.

### 4 Long-term summer and winter trends of sulfate in the Elbrus ice

From the winter/summer dissection made on the basis of the ammonium and succinate stratigraphy (Sect. 2), monthly means as well as half-year summer and winter means were calculated over the 1774 to 2010 period. In Figure 6, we report the seasonal cycle of sulfate, ammonium, and succinate averaged across a pre-industrial period (1775-1825 AD) and two different periods of the industrial period (1940-1960 and 1980-2000 AD). Individual sulfate half-year summer and winter means are reported in Figure 7, considering raw sulfate data and those regarded as free of dust ($SO_4^{2-}_{red.}$), respectively.

A few outlier of unknown origin were observed in the sulfate raw data set including 22 ppm at 166.65 m depth (summer 1780 AD), 2248 ppb at 146.38 m depth (summer 1862 AD), 1080 ppb at 146.11 m depth (summer 1863 AD), and 864 ppb at 154.73 m depth (winter 1833/34 AD). These individual values were removed when calculating the corresponding half-year





summer and winter means reported in Figure 7. In addition, single winter samples with sulfate levels of 815 ppb at 160.62 m depth (winter 1810/11 AD) and 3 data points from 150 to 230 ppb corresponding to winters 1786-87, 1827-28, and 1844-45 were not considered and corresponding half-year winter values were not reported.

A few ELB snow and ice layers are impacted by known volcanic eruptions. As discussed by Mikhalenko et al. (2015), ice
layers dated to 1911 and 1913 were probably impacted by the 1912 AD Katmai eruption, and summer layers of 1836 and 1837 by the 1835 AD Coseguina eruption, respectively. In addition, we suspect the 1854 AD Shiveluch eruption to have impacted summer 1854 ice layer and finally, although less evident since this part of the core is made up of splitted ice (see Section 2), the Cotopaxi 1877 AD eruption may have influenced the winter 1877/1878 layer. To discuss the long-term trends of sulfate in relation to growing $SO_2$ emissions, these half-year summer and winters means suspected to be contain volcanic
debris, were discarded in Figure 8. To minimize the effect of year-to-year variability due to meteorological transport conditions we added the first component of single spectra analysis (SSA) with a five-year time window in Figure 8, for the total sulfate data and those which are considered to be free of dust input ($SO_4^{2-}{}_{red.}$ values), respectively.

The dust influence on the long-term SSA winter trend is rather insignificant and if existing (i.e. effect of < 10 ppb) remaining limited to two decades around 1870 and the recent decade (2000-2010) (Fig. 8). Remaining negligible prior to 1850, the dust
effect on the summer trend gradually increases after 1950, reaching often 100 ppb after 1960. This change results from the long-term increasing trend of calcium summer concentrations (from $74 \pm 24$ ppb prior 1850 to $370 \pm 193$ ppb between 1960 and 2010), resulting from changes in precipitation and soil moisture content in Levant region (Syria and Iraq) and occurrence of drought in North Africa and Middle East regions (Kutuzov et al., this issue).

As seen in Figure 8, the mean summer and winter pre-industrial sulfate levels in ELB ice (taken as the mean value observed
from 1774 to 1850) is of 116 ppb and 69 ppb (113 ppb and 68 ppb, respectively for $SO_4^{2-}{}_{red.}$ values). In summer as in winter, the $SO_4^{2-}{}_{red.}$ values remained close to the pre-industrial values until 1910-1920 (125 ppb instead of 113 ppb in summer, 66 ppb in winter). After 1920, sulfate levels increased at a mean rate of 6 ppb per year (5.7 ppb yr$^{-1}$ for $SO_4^{2-}{}_{red.}$) in summer, and with 1.5 ppb yr$^{-1}$ for $SO_4^{2-}$ and $SO_4^{2-}{}_{red.}$ in winter. The sulfate increase then accelerated between 1950 and 1975 (11 ppb yr$^{-1}$, 10.5 ppb yr$^{-1}$ for $SO_4^{2-}{}_{red.}$), until a maximum of 730 ppb (663 ppb of $SO_4^{2-}{}_{red.}$) was reached between 1980 and 1990. After
1990, sulfate levels decreased to 590 ppb (490 ppb of $SO_4^{2-}{}_{red.}$) during the first decade of the twenty first century. In winter, the sulfate increase accelerated between 1950 and 1975 (4.5 ppb yr$^{-1}$ for $SO_4^{2-}$ and $SO_4^{2-}{}_{red.}$), to reach a maximum of 267 ppb (267 ppb of $SO_4^{2-}{}_{red.}$) between 1980 and 1990, followed by a subsequent decrease to 208 ppb (190 ppb of $SO_4^{2-}{}_{red.}$) in the first decade of the twenty first century.

## 5 Comparison between Elbrus and Alpine long-term sulfate trends

### 5.1 The Alpine CDD ice core Sulfate Records

The ELB sulfate long-term trend is compared with those previously extracted from the Alpine CDD site (ice cores denoted C10 and CDK in Figure 9). C10 sulfate data were presented in Preunkert et al. (2001), and those form CDK in (Legrand et


al., 2013). Since winter data from CDD are more limited (only a few pure winter layers are available between 1890 and 1930, Legrand et al., (2018) we here focus on the comparison of summer levels. The two CDD cores were dated by annual layer counting using the pronounced seasonal variations of ammonium. The two chronologies were in excellent agreement over their overlapping period from 1925-1990 (Legrand et al., 2013; Preunkert et al., 2000). A re-evaluation of the C10

chronology based on very recently made continuous measurements of heavy metals, as well as a comparison to a well-dated Greenland ice core record (McConnell and Edwards, 2008), resulted in a revised C10 chronology (Legrand et al., 2018). As for C10, continuous measurements of heavy metals are also available in the lowest part of CDK (Preunkert et al., in press). It was thus possible to identify the distinct Greenland increases of thallium, lead, and cadmium associated with the widespread start of coal burning at the beginning of the Industrial Revolution in 1890 CE also in the CDK core (at 117.8 m (90.5 mwe)).

This time marker was then used to constrain a revised annual layer counting in the early 20th-century part of the CDK record. In the following we compare summer trends of both Alpine CDD cores with the ELB ice core record, considering long term sulfate trends regarded to be free of dust influence (i.e., the $SO_4^{2-}{}_{red.}$ values).

**5.2 ELB versus CDD sulfate trends**

For summer (see Figure 9 a and b), the pre-industrial sulfate ELB value ($SO_4^{2-}{}_{red.}$ = 113 ppb) thus exceeded the CDD one (66

ppb) (Preunkert et al., 2000). A similar difference is observed for winter with $SO_4^{2-}{}_{red.}$ close to 68 ppb at ELB (Figure 7) compared to 20 ppb observed by Preunkert et al. (2000) at CDD. It is out of the scope of this work to discuss the cause of this difference between the two ice cores but we can first mention the existence of local volcanic sulfur emissions (as evidenced by direct on site observations of a sulfur smell nearby the ELB drill site). The pre-industrial summer level of dust free calcium samples at ELB (74 ppb, Kutuzov et al., this issue) is higher than the one at CDD (45 ppb, Legrand, 2002). That

may also contribute to the ELB/CDD difference of the sulfate pre-industrial level. Clearly, more work, including simulations with transport and chemistry models considering also oceanic emissions of DMS may help here.

Figure 9 compares the increasing summer sulfate trends of the ELB and CDD sites. Three major differences between the two sites are revealed: (1) an impact of anthropogenic emissions already significant in 1910 at CDD and not at ELB, (2) a maximum of the anthropogenic perturbation from 1970 to 1980 at CDD and 10 years after (1980-1990) at ELB, and (3) a

less pronounced re-decrease at the beginning of the 21th century at ELB compared to CDD.

As discussed by Kutuzov et al. (this issue), 10 day backward air mass trajectories calculated for the ELB site using the NOAA HYSPLIT-4 model suggest that, in summer, air masses arriving at ELB mainly originate from the nearby Georgia, Azerbaijan, Syria, Irak, and from Turkey, South Russian, and North of Iran. We report in Figure 10 emissions of $SO_2$ from these countries and from a few others located further north (Ukraine) and west (Bulgaria). In these countries $SO_2$ emissions

reached maximum in the late 80's or later (for Turkey and Iran). This feature clearly differs from the situation at CDD where countries around the site (France, Italy, Spain, Switzerland and Germany), thought to be the main contributors for sulfate in CDD ice (Fagerli et al., 2007), exhibit a maximum between the early 70's and the early 80's (Figure 10).


On this basis and as a first attempt, we compare the ELB and CDD summer sulfate trends with $SO_2$ emissions from surrounding countries. It can be seen that the impact of growing anthropogenic $SO_2$ emissions started later at ELB (after 1920) compared to CDD (after 1900). The 10-year delay of the sulfate maximum at ELB compared to CDD is also well seen in the enhancement course of $SO_2$ emissions. Note also that as indicated by the emissions, the maximum enhancement at

ELB (550 ppb between 1980 and 1990) is slightly weaker that the one at CDD (665 ppb between 1974 and 1984) (Table 3). Finally, consistently with $SO_2$ emission changes, the recent sulfate decrease is more pronounced at CDD than ELB with a recovered 2005 level (254 ppb) close to the 1950 one (234 ppb). At the ELB site this is not the case, here the 2005 level (380 ppb in 2005) is found to be still around two times higher than the one of 1950 (180 ppb in 1950).

**6 Conclusions**

Based on the ammonium and succinate stratigraphy, the upper 168.6 m of the deep ice core extracted at Mt Elbrus (Caucasus) in 2009 were dated by counting annual layers back to 1774 CE. The derived seasonally resolved chemical records cover the years 1774-2009 making this ice core particularly useful to reconstruct many aspects of atmospheric pollution in central Europe from pre-industrial times (1850 CE) to present-day. Below 169 m depth the annual counting is not possible but radiocarbon analysis of the particulate organic carbon fraction in the basal ice of the glacier suggests an age

of ~1670 ± 400 cal yr BP. We have examined the impact on the chemical composition of the Elbrus ice layers of arrival at the site of large dust plumes originating from Sahara and Middle East. We then report on the sulfur pollution. The ELB sulfate record indicates a four- and six-fold increase from prior to 1900 to 1980-1995 in winter and summer, respectively. Still moderate at the beginning of the 20[th] century, the sulfate increase accelerated after 1950, levels reaching a maximum in 1980-1990 (730 ± 152 ppb in summer) and subsequently decreasing to 630 ± 130 ppb in summer at the beginning of the 21[th]

century. These long-term sulfate changes observed in the ELB ice cores are compared with those previously obtained in Alpine ice. Consistently with past $SO_2$ emission inventories, a more pronounced decrease of the sulfur pollution over the three last decades is observed in western than central Europe.

**Data availability**

Sulfate and calcium data can be made available for scientific purposes upon request to the authors (contact:

suzanne.Preunkert@univ-grenoble-alpes.fr or michel.legrand@univ-grenoble-alpes).



**Author contributions**

S. Preunkert and M. Legrand performed research, analyzed ice samples and data, and wrote the original manuscript. S. Kutuzov performed research, analyzed data, and commented original manuscript. P. Ginot and V. Mikhalenko performed analysis and commented original manuscript. R. Friedrich analyzed ice samples and commented original manuscript.

5 **Acknowledgments.**

The study was supported by the RSF grant 17-17-01270. The « Les Enveloppes Fluides et l'Environnement- Chimie Atmosphérique » (CNRS) program entitled "Evolution séculaire de la charge et composition de l'aérosol organique au dessus de l'Europe" (1262c008 –ESCCARGO) provided funding for ion chromatography analysis, with the support of Agence de l'Environnement et de la Maîtrise de l'Energie, and the LIA Vostok provided funding for PO$^{14}$C measurements,
10 respectively. Thanks a lot to Simon Escalle for preparing the blank ice for the PO$^{14}$C measurements.



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


**Table 1.** Overview of masses (corrected for blanks but not for combustion efficiency) and conventional $^{14}$C ages of the Elbrus ice core samples combusted in the REFILOX system. Calibrated date ranges are shown at 68.2% confidence level and are rounded ac cording to (Millard, 2014).

| Sample name | Depth | Ice mass [g] | POC mass corrected [μgC] | $^{14}$C corrected [F$^{14}$C] | Calibrated 14C date BCE/CE at 68.2% | Calibrated $^{14}$C-age* range at 68.2% [yr cal BP] | Calibrated $^{14}$C-age* [yr cal BP] mean |
|---|---|---|---|---|---|---|---|
| ELB-176-03 | 177.11 ± 0.22 m (137.89 ± 0.18 mwe) | 295 | 4.5±0.5 | 0.914± 0.043 | 670 CE – 1245 CE | 1280 - 705 | 1040 |
| ELB-178-03 | 179.19 ± 0.14 m (139.59 ± 0.12 mwe) | 300 | 5.6±0.5 | 0.955± 0.098 | 130 CE – 770 CE | 1820 - 1180 | 1530 |
| ELB-181-01 | 181.50 ±0.13 m (141.19 ± 0.11 mwe) | 260 | 4.4±0.5 | 0.932± 0.020 | 440 CE – 1290 CE | 1510 - 660 | 1110 |
| ELB-181-03 | 182.02 ± 0.13 m (141.62 ± 0.11 mwe) | 320 | 6.5±0.5 | 0.875± 0.021 | 90 BCE – 680 CE | 2040 - 1270 | 1670 |

5   * age before 1950 CE



**Table 2.** Composition of aerosol collected at Erdemli during dust events from Middle East and at Heraklion during dust events from Sahara (from Koçak et al., 2012). The magnesium sea-salt contributions were calculated via the $Na^+$ content, after having subtracted its dust contribution calculated as 0.08 times the $Ca^{2+}$ content (see text).

| Species | Erdemli (October 2007) | Heraklion (April 2008) |
|---|---|---|
| $[Na^+]$ | 1148 ng m$^{-3}$ (sea-salt: 581 ng m$^{-3}$) | 1106 ng m$^{-3}$ (sea-salt: 445 ng m$^{-3}$) |
| $[Mg^{2+}]$ | 423 ng m$^{-3}$ (sea-salt: 70 ng m$^{-3}$) | 407 ng m$^{-3}$ (sea-salt: 53 ng m$^{-3}$) |
| $[Ca^{2+}]$ | 7085 ng m$^{-3}$ | 8264 ng m$^{-3}$ |
| $[Mg^{2+}]/[Ca^{2+}]$ in dust | 0.050 | 0.043 |



**Table 3.** Mean chemical composition of snow layers deposited in periods characterized by low and high dust inputs (1974-1978 and 1996-2000, respectively). Values in parenthesis are mean values calculated after removal of samples containing dust (denoted red.). Δ refer to the difference between total and dust reduced (red.) values (i.e., [X] - [X$_{red.}$] for the X species).

|  | 1974-1978 AD | 1996-2000 AD |
|---|---|---|
| $Ca^{2+}$ ($Ca^{2+}_{red.}$) | 286 ppb (169 ppb) Δ = 117 ppb | 544 ppb (186 ppb) Δ = 358 ppb |
| $Mg^{2+}$ ($Mg^{2+}_{red.}$) | 25.0 ppb (19.5 ppb) Δ = 5.5 ppb | 29.5 ppb (18.0 ppb) Δ = 11.5 ppb |
| $K^+$ ($K^+_{red.}$) | 24 ppb (20 ppb) Δ = 4 ppb | 30 ppb (20 ppb) Δ = 10 ppb |
| $Na^+$ ($Na^+_{red.}$) | 31 ppb (25 ppb) Δ = 6 ppb | 40 ppb (24 ppb) Δ = 16 ppb |
| $Cl^-$ ($Cl^-_{red.}$) | 67 ppb (55 ppb) Δ = 12 ppb | 88 ppb (46 ppb) Δ = 22 ppb |
| $NH_4^+$ ($NH_4^+_{red.}$) | 177 ppb (165 ppb) Δ = 12 ppb | 199 ppb (149 ppb) Δ = 50 ppb |
| $NO_3^-$ ($NO_3^-_{red.}$) | 292 ppb (279 ppb) Δ = 13 ppb | 436 ppb (310 ppb) Δ = 126 ppb |
| $SO_4^{2-}$ ($SO_4^{2-}_{red.}$) | 649 ppb (598 ppb) Δ = 51 ppb | 738 ppb (550 ppb) Δ = 188 ppb |
| $HCOO^-$ ($HCOO^-_{red.}$) | 168 ppb (135 ppb) Δ = 33 ppb | 184 ppb (113 ppb) Δ = 71 ppb |
| $CH_3COO^-$ ($CH_3COO^-_{red.}$) | 48 ppb (37 ppb) Δ = 11 ppb | 49 ppb (39 ppb) Δ = 10 ppb |





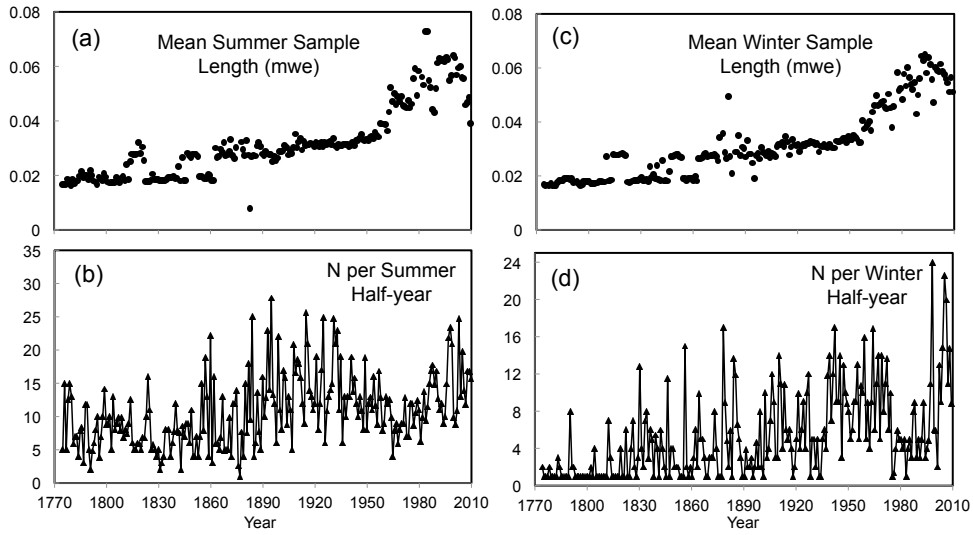

**Figure 1:** (a) and (c): Mean length of individual samples in summer and winter layers (in mwe). (b) and (d): Numbers of samples (N) spanning summer and winter half-years.

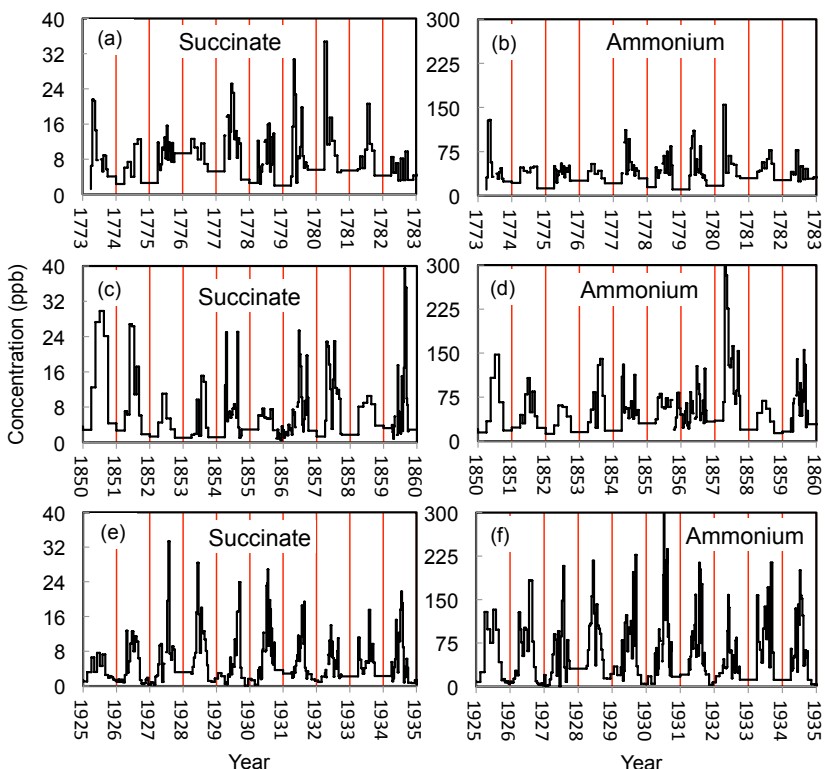

**Figure 2.** ELB ice chronology at depth intervals of 166.2 to 168.5 m (top), 154.4 to 156.5 m (mid), and from 99.8 to 107.3 m (bottom), based on the ammonium and succinate stratigraphy. Vertical red lines denote yearly dissection based on identification of winter layers (see Sect. 2.2). For the two oldest time-periods (1773-1782 and 1850-1859), each sample was 2 cm long whereas for the most recent time period (1925-1934) one sample was on average 4 cm long.

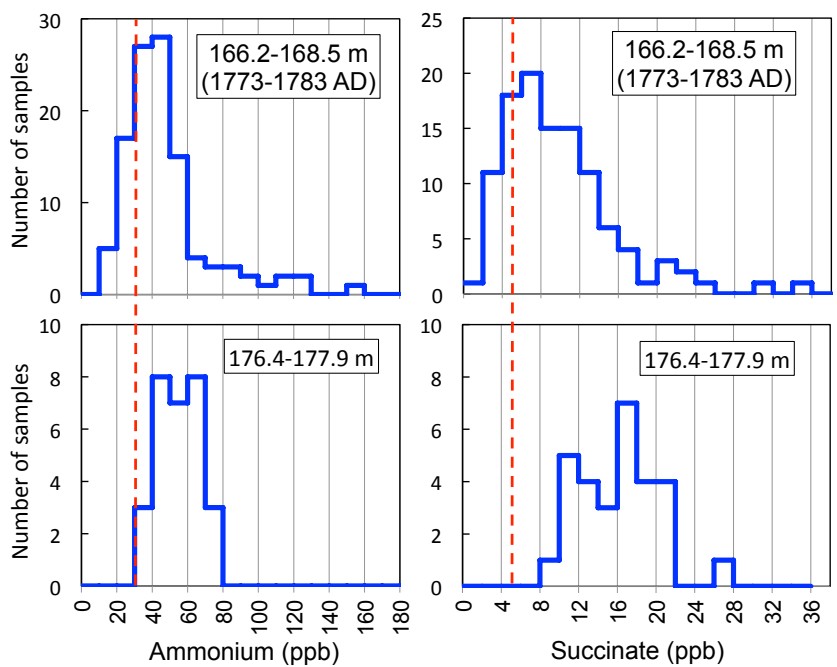

**Figure 3.** Distribution of succinate and ammonium concentrations observed in the deepest ice layers for which annual counting was possible (i.e., above 168.5 m depth, top) and 10 m below (bottom). The vertical red dashed bars denote the values of the winter criteria.

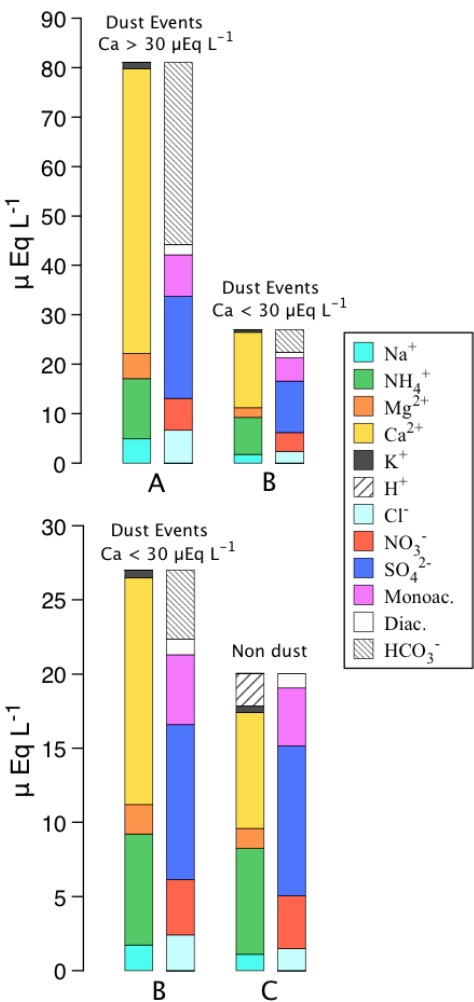

**Figure 4.** Mean ionic content of ELB layers deposited between 1950 and 1980. A and B: Mean composition of dust event samples containing more and less than 600 ppb (i.e., 30 $\mu$Eq L$^{-1}$) of calcium. B and C: Mean composition of dust event samples containing less than 600 ppb of calcium compared to samples free of dust. Abbreviations Monoac. and Diac. stand for $C_1$-$C_3$ monocarboxylates and $C_2$-$C_5$ dicarboxylates, respectively (see Eqs. 2 and 3 in Sect. 3).





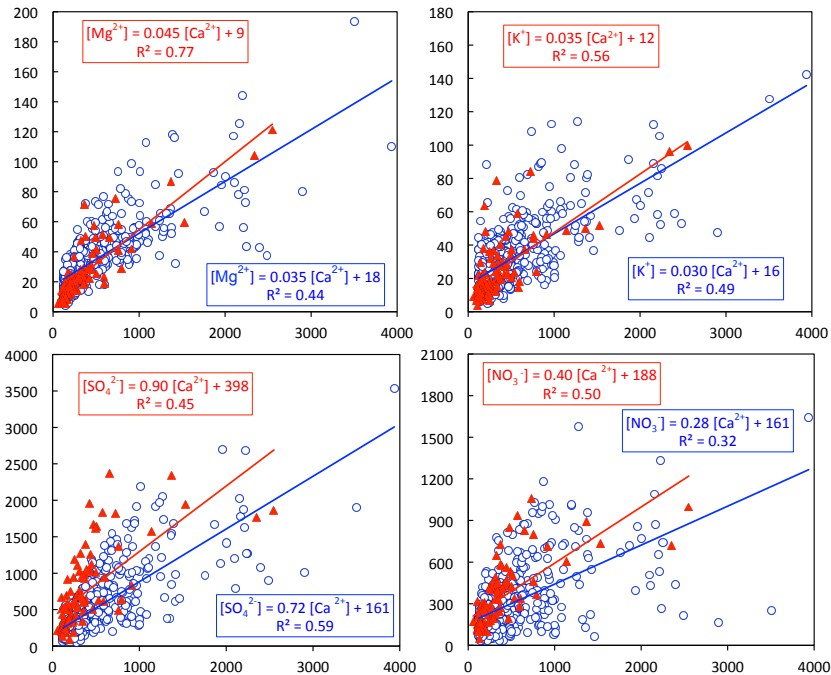

**Figure 5.** Comparison of the chemical compositions (magnesium, potassium, sulfate, and nitrate versus calcium) of dust ice samples from Col du Dome (CDD, in red) and Elbrus (in blue). Data from CDD are from Preunkert (2001). All concentrations are in ppb.





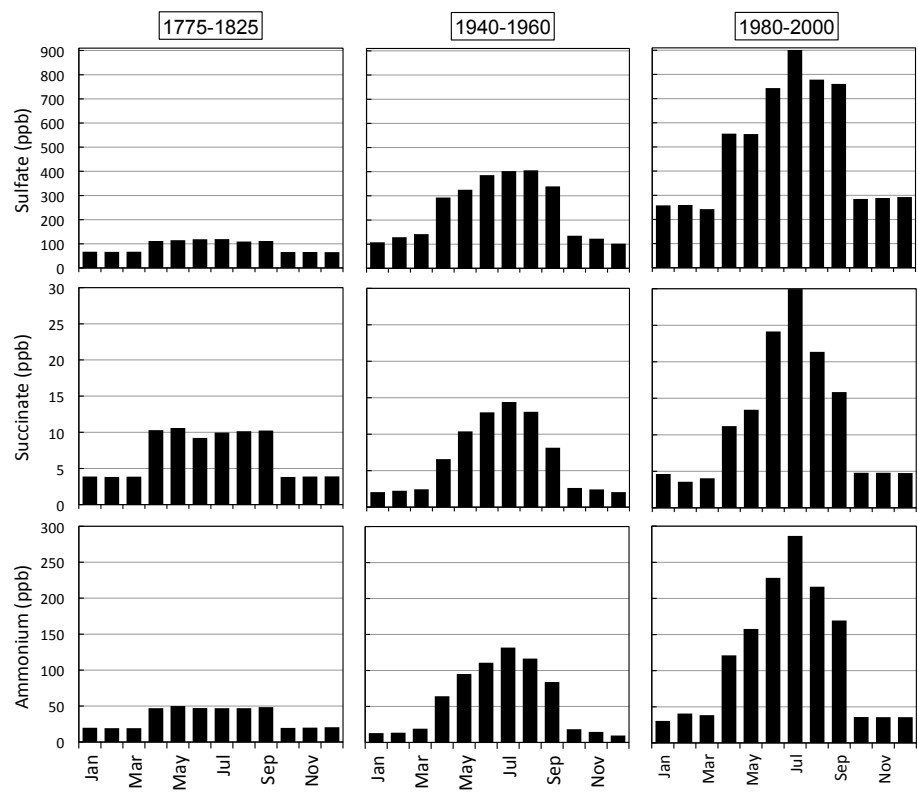

5    **Figure 6.** Monthly means of sulfate, succinate, and ammonium (used for seasonal dissection, see text), over three different time periods.

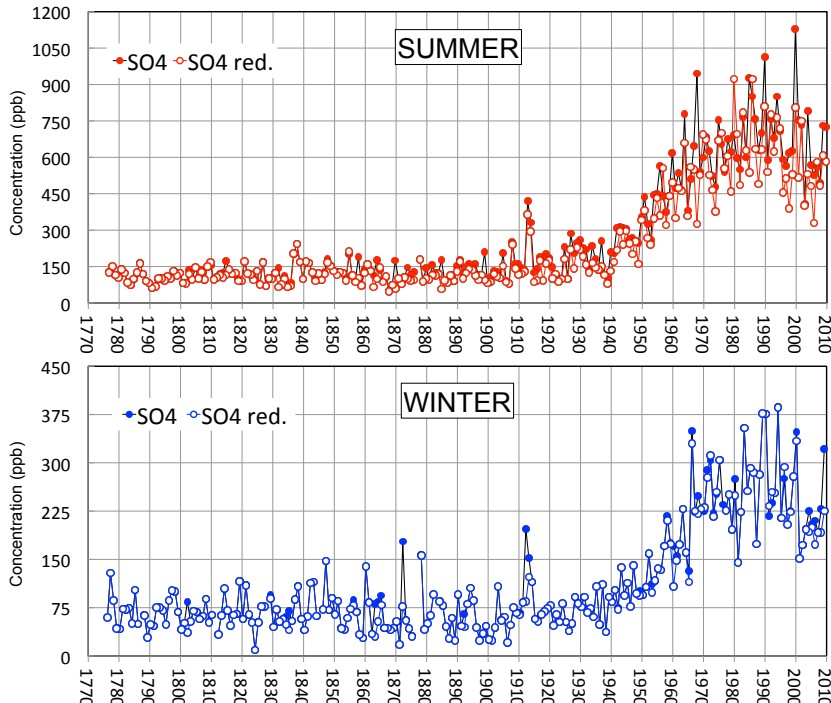

5   **Figure 7.** Individual summer (red) and winter (blue) half-year means of sulfate along the ELB ice core. Solid circles refer to values calculated considering all samples. Open circle data (SO4 red.) were calculated after having removed samples considered to be impacted by dust events (see Sect. 4).

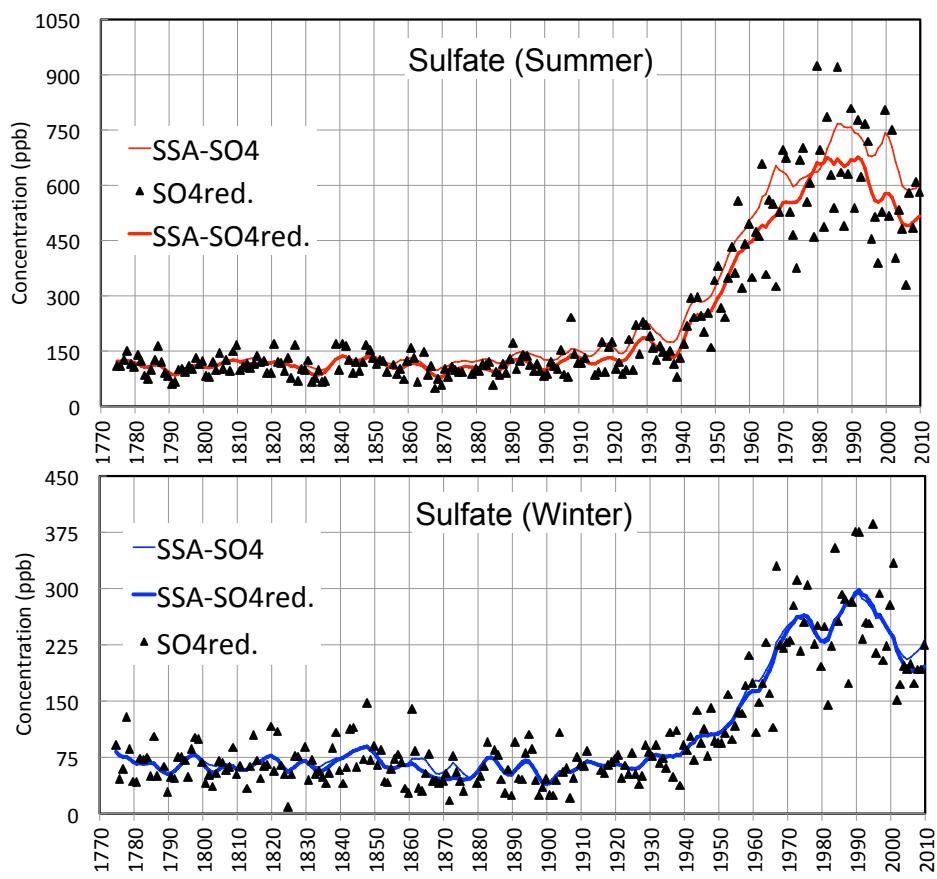

5   **Figure 8.** ELB half-year sulfate summer and winter means from 1774 to 2010 (triangles) calculated after removal of dust samples (SO4$_{red.}$). The solid thick lines (blue for winter, red for summer) are the first SSA component with a five-year time window (see section 6) calculated for the dust free considered dataset. The solid thin lines (blue for winter, red for summer) represent the first SSA component (five-year time window) when dust events were not removed from calculations. Samples suspected to be impacted by volcanoes were removed.



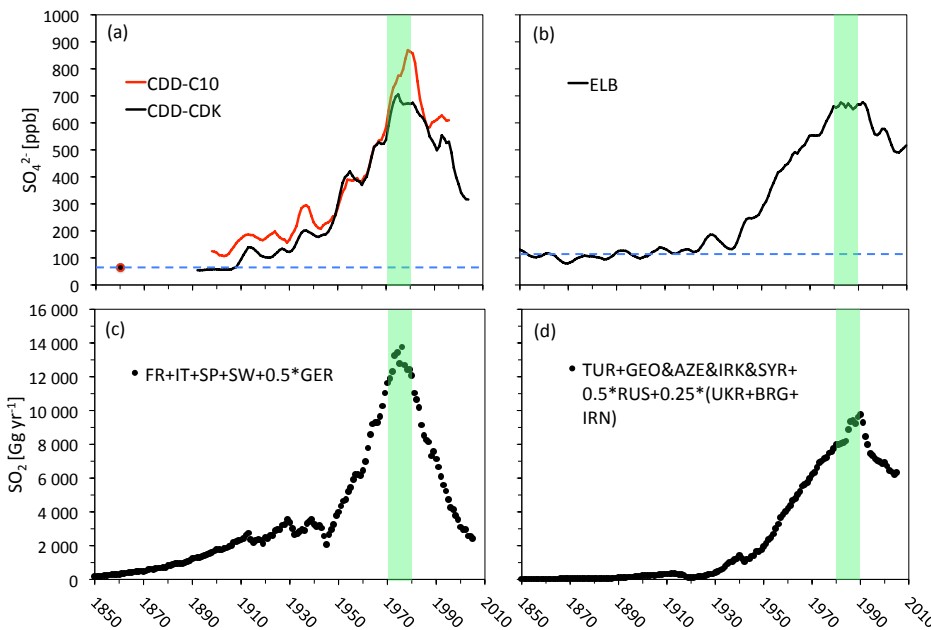

**Figure 9.** Comparison of the dust corrected long-term sulfate summer trends ($SO_4^{2-}{}_{red.}$ values) from two CDD ice cores (C10 and CDK) (a) and the ELB ice core (b) with $SO_2$ emissions from countries suspected to contribute to sulfate depositions at the two sites (c for CDD, d for ELB, see discussions in Section 5). (a) and (b): solid lines refer to the first SSA component with a five-year time window. Dashed blue lines refer to the respective pre-industrial sulfate levels (see section 5). (c): $SO_2$ emissions from France, Italy, Spain, Switzerland and half from Germany. (d): $SO_2$ emissions from Turkey, Georgia and Azerbaijan, half from Russia, a quarter from Ukraine, Bulgaria, and Iran. Green areas indicate the decades in which the sulfate levels reached their maxima.

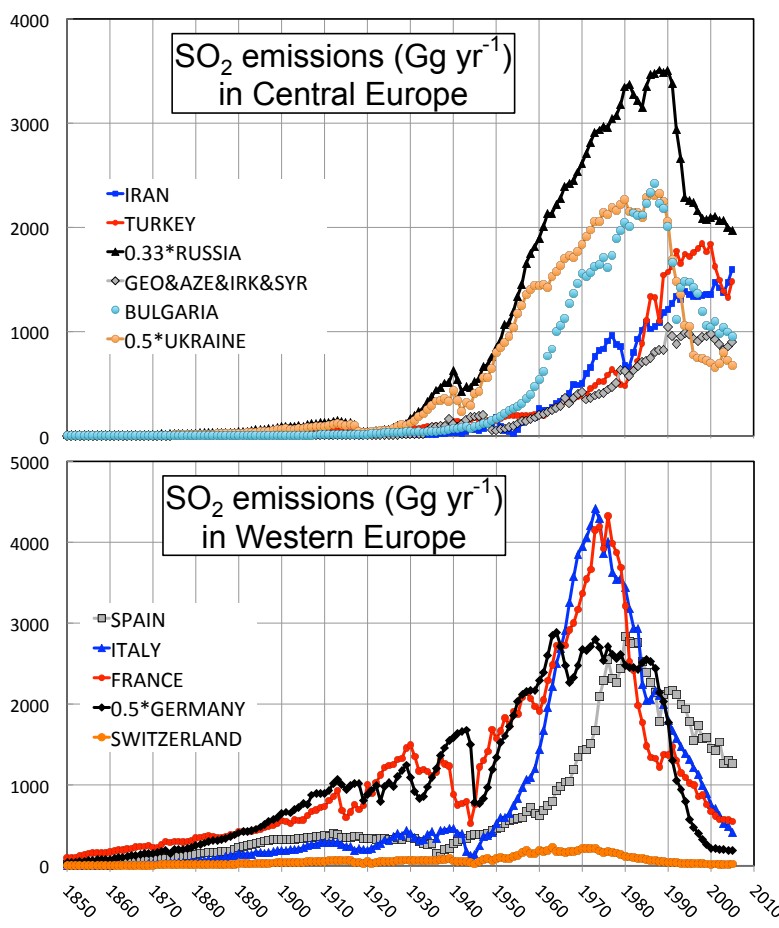

**Figure 10.** SO$_2$ emissions (from 1850 to 2005) from various countries located around the Caucasus (top) and the Alps (bottom). GEO&AZE&IRK&SYR denotes emissions from Georgia, Azerbaidjan, Irak, and Syria. Data are from Smith et al. (2011).