# Peer review of "The Elbrus (Caucasus, Russia) sulfate ice core record: reconstruction of past anthropogenic sulfur emissions in southeastern Europe"

_Atmospheric Chemistry and Physics, 2019_

## Referee Comment (RC1) · Anonymous Referee #2 · 19 Jun 2019

This publication reads like a solid piece of work, well written, and logically structured. The caveat is that I am not an icecore specialist- and if there are methodological issues in this part, I have probably not spotted that. From a general atmospheric chemistry perspective, however, the manuscript and story make a lot of sense. I can therefore recommend this manuscript for publication in ACP, with some minor suggestions for improvements below.

Minor suggestions: General: As this manuscript is submitted to a more general Atmospheric Chemistry journal, I would recommend to spell out/explain specialized abbreviations used in this manuscript. E.g. I didn't know the meaning of Yr cal BP; also BP,

[Figure]

CE may not be known to all readers. Possibly a table ?

General: it would be useful if in addition to concentrations also the deposition fluxes would be presented, which is the more obvious quantity for comparison with models.

P1 l. 19 focus on dust-free sulfur pollution. (to clarify).

P1 l. 26 I would say also the much later onset is an important piece of information, which confirms knowledge on industrialization.

P2 l 2 In general short lived climate forcers, with one of the most important components being aerosol.

P 2 l 6 this is somewhat naïve statement, as models will usually calculate the concentrations and verify them with observations. Only from the satellite era onward, aerosol is assimilated but not in 'climate' models.

P 2 l 6 A number of other continental icecores are mentioned, but only CDD is explored later in the text. It is not entirely clear, why a comparison with the other icecores is not included in the manuscript.

2 l 26 Another argument is that there is a quite strong seasonal dependency of the oxidation chemistry of SO2, which has probably been oxidant limited in the emission era.

P3 l. 15 explain meter water equivalent, and if this information is available how do these precipitation rates compare to a larger footprint around Mt. Elburus?

P 3 l 28-32. Later in the text outliers are removed, are these outliers related to these known problems? If not what could be the cause of such outliers?

P 4 l. 12 Again for non experts explain whether the decrease of NH4 with depth is a 'real' signal, or rather related to gradual degradation/oxidation with time.

P 4 l. 34 I understand the chemical stratification is a preferred method compared to

radio carbon dating, can you confirm because that is because of higher accuracy?

P 6 l. 27 the 616 and 67 numbers are the samples influenced by high dust? Sentence is ambiguous.

P 7 l. 4 at the best=>at the most

P 7 l. 14 Dust may contain a quite large fraction of CaSO4, which is quite insoluble under alkaline conditions, may be dissolve when more acidic. If I understood well this would not be picked up in the analysis, and can not influence the trend estimates? Please confirm.

P 8 l15- you can mention here that the corrected values were rather consnstant as also shown in Figure 9. p. 9 l 1 Please provide some plausible reasons for the outliers, or connect to the statements in the analysis section.

P 9 l. 27 It would be good to mention here which emission database was used.

---

## Referee Comment (RC2) · Margit Schwikowski (Referee) · 28 Jun 2019

This manuscript presents an ice core record of sulfate from a glacier on Mount Elbrus covering the time period 1774-2009. Generally, it is well written and structured and mostly scientifically sound (see comments below). The high-quality data set fills a gap, since it is the first sulfate record from South-Eastern Europe. Regional data on pre-industrial to industrial concentration changes of major aerosol components are essential to constrain emission estimates used in modelling the aerosol effect on climate. I therefore expect that this record will have an impact. The manuscript definitely deserves publication, after taking into account the comments and suggestions listed in

the following.

Specific comments:

There is very little information about the ice core itself. The coordinates are just given in the abstract and the length in the introduction. I suggest adding a short paragraph about the Elbrus ice core, including some additional information, e.g. name of the glacier, ice thickness, ice temperature, and net accumulation rate. In the abstract it is called a deep ice core, but that is relative. More important is if it reached bedrock or not. In addition, it would also be good to summarize briefly previous work published on this core.

Use the term South-Eastern Europe instead of Central Europe for the source area of emissions detected in the Elbrus record.

You observe a stronger thinning with depth of the winter layers compared to the summer layer. This is interesting. Is it due to a change of precipitation seasonality or is it an artefact caused by diffusion of chemical tracers or even different flow behaviour of summer and winter layers?

Dating of the core: This is so central for the interpretation and it was extended compared to the previous publication (Mikhalenko et al., 2015). I therefore suggest including a depth-age figure with the 14C dating points to give an idea about the thinning (can this be fitted with a glaciological flow model?). Also the volcanic horizons used to anchor the counted layers should be shown. You evoke basal melting to explain why the deepest ice is so young. Does this mean, the glacier is not frozen to bedrock at the drilling site? This has implications on the thinning. Please clarify.

I am not convinced by the equidistant binning of the summer and winter layers to obtain monthly values. This requires the absence of seasonality in precipitation and snow preservation. Precipitation data from nearest meteorological stations show strong seasonality (Kozachek et al., 2017). Since the monthly data are not really discussed,

accept for showing the seasonality of chemical tracers in Fig. 6, I suggest deleting this part and the figure.

Identification of annual layers and attribution of summer and winter layers: You use two criteria for that (ammonium and succinate). To which of the two do you give priority when the two signals do not agree? How does the attribution of summer and winter layers presented in this manuscript agree with the one based on the stable isotope record of the same core (Kozachek et al., 2017)?

14C-dating: Was the AMS equipped with a gas ion source? You used a rather old version of Oxcal. I suggest using an updated version.

Ion balance: Use the same unit (either ppb or uEq/L) in the text and in Figure 4.

Attribution of dust sources: This part of the manuscript is not convincing to me. What is the argument to relate high Ca concentrations to Saharan dust and low Ca concentrations to sources in the Middle East? The plots in Figure 5 show a large scatter and low correlation coefficients, so I wonder if the ion ratios you discuss are significantly different. For the ions with strong anthropogenic influence this correlation analysis is anyway not meaningful without splitting the data set in the pre-industrial and industrial periods. To me this part of the manuscript is weak, distracts from the main message, and could be omitted. Important is to estimate the amount of sulfate originating from dust and correct for that when discussing anthropogenic sulfate.

Attribution of sulfate related to mineral dust: Instead of arbitrarily introducing a Ca level to identify dust events, I propose to look at the pre-industrial Ca to sulfate correlation. If both are highly correlated, you can use this ratio to correct for mineral dust sulfate in the industrial period. I recommend adding a map with the Elbrus site, outlining the dust and SO2 emission source areas.

Table 5 is mentioned in the text, but does not exist.

Discussion of outliers: This is hard to follow without seeing the raw data (which should

be shown anyway). Can some of the outliers be explained by volcanic events? It is strange that you don't see a signal of the largest eruption in the last centuries (Tambora, 1815) and the largest eruption in the Northern Hemisphere in the last centuries (Laki, 1783).

Comparison with emission estimates: You stress in the manuscript the importance to distinguish between summer and winter sulfate values and trends (to me the trends look similar). And then you compare this with emission estimates, which are annual values (I guess). This is inconsistent. You need to include the total anthropogenic sulfate record, which would also be very valuable for comparison with data sets from other ice cores, which are not resolved in summer and winter values. In addition, you give the impression that $SO_2$ emissions in winter are much lower than in summer. The opposite is the case. The major factor producing the difference in summer and winter values at high-alpine sites is the reduced vertical atmospheric transport in winter (and not the variation in source area). You need to explain this in the manuscript.

Considering the $SO_2$ emission source areas you identified it is strange that you just compare the Elbrus record with the CDD record from the Alps. I strongly recommend to include the sulphate record from Eastern Europe (from Belukha ice core, Eichler et al., ES&T 2012).

Figure 1. I don't see the point of showing the mean summer and winter sample length. This should not be so different from the sample resolution.

Technical corrections

Title: seems too long and a bit cumbersome. Suggestion: Reconstruction of anthropogenic sulfate trends from Elbrus ice core, Caucasus.

Abstract L. 18: After having examined. . . Rephrase and give the results: dust contribution to sulfate concentrations was identified and subtracted to focus on anthropogenic sulphate (not sulfur)

P2L4: Replace Andreae et al., 2015 with a newer estimate e.g. from IPCC.

P2L8: "impact" instead of "disturb"

P2L13-15: The Altai and Kamchatka are not part of Europe.

P2L16-19: ice cores have been investigated . . .to examine

P3L12: Give more details how ice cores were decontaminated (by removing xx cm from the outside of the core. . .)

P3L17: loss

P3L29: Give details, which fluid was used.

P4L7: For the ammonium seasonality earlier work should be cited (Maupetit et al., Atmos. Environ., 1995; Eichler et al., JGlac., 2000)

P6L14-15: Replace "disturb the chemistry" by changes the chemical composition

Table 1: Include 14C lab sample reference number.

---

## Author Response (AR1)

**Answer to Anonymous Referee #2**

*We would like at first thank the reviewer for his comments. In the following our answers are made in italic.*

This publication reads like a solid piece of work, well written, and logically structured. The caveat is that I am not an ice core specialist- and if there are methodological issues in this part, I have probably not spotted that. From a general atmospheric chemistry perspective, however, the manuscript and story make a lot of sense. I can therefore recommend this manuscript for publication in ACP, with some minor suggestions for improvements below.
Minor suggestions: General: As this manuscript is submitted to a more general Atmospheric Chemistry journal, I would recommend to spell out/explain specialized abbreviations used in this manuscript. E.g. I didn't know the meaning of Yr cal BP; also BP, CE may not be known to all readers. Possibly a table ?
*We agree and we have now specified in the "Basal ice Dating" section that: "As seen in Table 1, the mean age of the ELB-178-03 sample (1530 yr cal BP, i.e. 1530 years before 1950)…….;"*
*Also in the first sentence of the abstract: "This study reports on the glaciochemistry of a deep ice core (182 m long) drilled in 2009 at Mount Elbrus in the Caucasus, Russia. Radiocarbon dating of the particulate organic carbon fraction in the ice suggests that the basal ice dates to 280 ± 400 yr CE (Common Era)."*

General: it would be useful if in addition to concentrations also the deposition fluxes would be presented, which is the more obvious quantity for comparison with models.
*We agree that the knowledge of deposition fluxes would be useful to compare with model simulations. Unfortunately, that is not an easy task. Indeed, what we estimate on the basis of the seasonal dissection (or annual counting) is the annual ice thickness. This annual ice thickness systematically decreases with depth due to ice flow. Ideally, using the annual ice thickness versus depth and a good ice flow model (that does not exist) it would be possible to derive the original accumulation rate. But even with that, you have to consider a possible strong erosion of snow by wind after deposition.*

P1 l. 19 focus on dust-free sulfur pollution. (to clarify).
*OK done here and throughout the text.*

P1 l. 26 I would say also the much later onset is an important piece of information, which confirms knowledge on industrialization.
*As now discussed when comparing the three sites (see also our answer to your comment on comparison with other ice cores) in section 5, we also emphasized the difference in the onset: "Long-term dust-free sulfate trends observed in the ELB ice cores are compared with those previously obtained in Alpine and Altai (Siberia) ice, the most important differences consist in a much earlier onset and a more pronounced decrease of the sulfur pollution over the three last decades in western Europe than south-eastern Europe and Siberia."*

P2 l 2 In general short lived climate forcers, with one of the most important components being aerosol.
*OK done*

P 2 l 6 this is somewhat naïve statement, as models will usually calculate the concentrations and verify them with observations. Only from the satellite era onward, aerosol is assimilated but not in 'climate' models.

*You are right and we now corrected the sentence as follows: "However, uncertainties still exist in quantifying the climatic impact of aerosols. The spatial distribution of aerosols is very heterogeneous and requires therefore numerous observations to make these parameters useful as inputs and constraints for transport and chemistry models."*

P 2 l 6 A number of other continental ice cores are mentioned, but only CDD is explored later in the text. It is not entirely clear, why a comparison with the other icecores is not included in the manuscript.

*Yes, in the draft the ELB record was only compared to the CDD (alpine) record. In the revised version we also add a comparison with the one extracted from the Belukha glacier in the Siberian Altai. The abstract was changed consistently.*

2 l 26 Another argument is that there is a quite strong seasonal dependency of the oxidation chemistry of SO2, which has probably been oxidant limited in the emission era.

*Yes you are right: three factors influence the seasonality: the upward transport intensity (maximum in summer) is the main factor, the second and third are the seasonality in the emission with a slight maximum in winter counteracted by the lowering of the conversion $SO_2/SO_4$.*

P3 l. 15 explain meter water equivalent, and if this information is available how do these precipitation rates compare to a larger footprint around Mt. Elbrus?

*Annual firn and ice (density varying from 0.3 to 0.9 $g\ cm^{-3}$) thickness are commonly converted into a water column. As answered above, the link between ice annual thickness and local precipitation rate is anyway quite complex. For your information, Kozachek et al. (Climate of the Past, 13, 1473-489, 2017) reported a mean annual ice thickness close to 1.3 mwe in the recent layers (see also the new Figure 2 in the revised version) against a precipitation rate of 1.7 m of water at the site of Klukhorskiy Pereval located at 2037 m elevation.*

P 3 l 28-32. Later in the text outliers are removed, are these outliers related to these known problems? If not what could be the cause of such outliers ?

*Generally speaking single values considered as outliers are very likely due to contamination during the subsampling or due to the poor quality of the ice. To better illustrate these outliers, we report them in Figure 7 together with all raw data.*

P 4 l. 12 Again for non experts explain whether the decrease of NH4 with depth is a 'real' signal, or rather related to gradual degradation/oxidation with time.

*No, as far as we know there is no evidence from numerous other ice core studies that the decrease of ammonium with depth is due to a degradation. More likely the increase of ammonium in the recent layer (as for nitrate) is related to atmospheric changes but a detail discussion of these trends is out of the scope of this paper that focuses on sulfate.*

P 4 l. 34 I understand the chemical stratification is a preferred method compared to radio carbon dating, can you confirm because that is because of higher accuracy?

*Yes, definitely when the annual counting is possible the accuracy is typically a few years for a period of 100 years. When not possible because of poor record in ice of the seasonal atmospheric contrast, and in the absence of absolute horizons such as volcanic events, the*

*unique possibility is the radiocarbon dating (which is far less accurate). In the revised version we report a dating figure (Figure 5) that illustrates fairly well this point.*

P 6 l. 27 the 616 and 67 numbers are the samples influenced by high dust? Sentence is ambiguous. *OK we reworded this sentence: "In this way, 616 (on a total of 2524) summer and 67 (on a total of 1150) winter samples were considered as containing large amount of dust."*

P 7 l. 4 at the best=>at the most :
*OK Done*

P 7 l. 14 Dust may contain a quite large fraction of CaSO4, which is quite insoluble under alkaline conditions, may be dissolve when more acidic. If I understood well this would not be picked up in the analysis, and can not influence the trend estimates? Please confirm.
*Fraction of gypsum (primarily emitted) may indeed be quite insoluble. However the other fraction coming from neutralization of calcium carbonate during transport (sulphuric acid or $SO_2$) would be more soluble. Also the change of acidity over time in this region is not so large than in the Alps since together with the increase of anthropogenic species (sulfate and nitrate) we have an increase of alkaline calcium (as discussed in the companion paper).*

P 8 l15- you can mention here that the corrected values were rather constant as also shown in Figure 9. ???
*We are not sure if we understand your question. This figure appears far later in the text.*

p. 9 l 1 Please provide some plausible reasons for the outliers, or connect to the statements in the analysis section.
*See our previous answer for outliers.*

P 9 l. 27 It would be good to mention here which emission database was used.
*Yes done: "Using data from Smith et al. (2011), available at http://sedac.ciesin.columbia.edu/data/set/haso2-anthro-sulfur-dioxide-emissions-1850-2005-v2-86, we report in Figure 12 emissions of SO2 from countries located nearby the ELB site: Georgia, Azerbaijan, Syria, Irak, Turkey, Russian, Iran or located further north (Ukraine) and west (Bulgaria)." And we also add to the reference list :" "Smith, SJ, J van Aardenne, Z Klimont, RJ Andres, A Volke, and S Delgado Arias. (2011). Anthropogenic Sulfur Dioxide Emissions: 1850–2005, Atmospheric Chemistry and Physics, 11:1101–1116."*

**Answer to Referee Margit Schwikowski**

*We would like at first thank the reviewer for her comments. In the following our answers are made in italic.*

This manuscript presents an ice core record of sulfate from a glacier on Mount Elbrus covering the time period 1774-2009. Generally, it is well written and structured and mostly scientifically sound (see comments below). The high-quality data set fills a gap, since it is the first sulfate record from South-Eastern Europe. Regional data on pre-industrial to industrial concentration changes of major aerosol components are essential to constrain emission estimates used in modelling the aerosol effect on climate. I therefore expect that this record will have an impact. The manuscript definitely deserves publication, after taking into account the comments and suggestions listed in the following.

Specific comments: There is very little information about the ice core itself. The coordinates are just given in the abstract and the length in the introduction. I suggest adding a short paragraph about the Elbrus ice core, including some additional information, e.g. name of the glacier, ice thickness, ice temperature, and net accumulation rate. In the abstract it is called a deep ice core, but that is relative. More important is if it reached bedrock or not. In addition, it would also be good to summarize briefly previous work published on this core.

*Thanks for this comment, we added a paragraph with more information on the drill site, the ice core and previous works done on this ice core, as follows "A deep ice core was drilled to bedrock (182.6 m, i.e. 142.1 meter water equivalent (mwe)) in 2009 on the western plateau of Mt. Elbrus (43°21'N, 42°26'E; 5115 m above sea level) in the Caucasus, Russia (Fig. 1). Glaciological settings of the drill site are detailed in Mikhalenko et al. (2015). In brief, the surface of the glacier plateau is about 0.5 km², and the surface snow accumulation at the drill site is about 1.5 mwe yr⁻¹. Ice-penetrating radar measurements made in 2007 and 2009 revealed a maximum glacier thickness of 255 ± 8 m at the central part of the plateau, and minimum values of ~60 m near the western border of the glacier. Borehole measurements indicated temperatures of -17°C at 10 m depth and -2.4°C at 181.8 m depth. Occasionally melting of surface snow can occur, however, the thickness of the infiltration ice layers, which do not form every year, does not exceed 10 mm. After the overall presentation from Mikhalenko et al. (2015), two other studies of the ELB ice core were dedicated to black carbon (Lim et al., 2017) and water stable isotope composition on the 126 m upper layers (Kozachek et al., 2017). "*

Use the term South-Eastern Europe instead of Central Europe for the source area of emissions detected in the Elbrus record.

*OK done through the text.*

You observe a stronger thinning with depth of the winter layers compared to the summer layer. This is interesting. Is it due to a change of precipitation seasonality or is it an artefact caused by diffusion of chemical tracers or even different flow behaviour of summer and winter layers?

*We assume that the reviewer refers here to Figure 1 (mean summer and winter sample length). The initial aim of this figure was to illustrate the fact that we adapted the depth resolution of the ice sampling (from 10 cm at the top to 2 cm at the bottom) to the decrease of annual thickness with depth, in view to minimize the lost of temporal resolution. In fact this aspect is particularly important in the companion paper that deals with changes of the frequency of sporadic dust events over time. In the revised version this figure has been reworked to illustrate the thinning of summer, winter, and annual layers with depth.*

*As now shown, indeed the winter layer thickness decreases more than the summer ones but only in the lowest part of the core (below 155 mwe, see Figure 1d). Already observed in*

*Alpine small-scale glaciers we assume that this effect is likely due to more wind erosion of winter than summer snow layers upstream the drill site. Note also that no changes in the seasonal precipitation contributions were observed at least over the last 100 years for which the seasonal precipitation was reconstituted for this ice core (Kozachek et al., 2017). The text in the manuscript in section 2.1. was revised accordingly. "The large decrease of the net snow accumulation in winter below 155 m (Fig. 2), likely due to more wind erosion of winter than summer snow layers upstream the drill site as already observed at other high altitude glacier sites (e.g. Preunkert et al., 2000), leads to a more pronounced loss of resolution in these winter layers compared to the surface layers (12 samples per winter near the surface and 1-2 samples per winter at 157 m depth)."*

Dating of the core: This is so central for the interpretation and it was extended compared to the previous publication (Mikhalenko et al., 2015). I therefore suggest including a depth-age figure with the 14C dating points to give an idea about the thinning (can this be fitted with a glaciological flow model?). Also the volcanic horizons used to anchor the counted layers should be shown. You evoke basal melting to explain why the deepest ice is so young. Does this mean, the glacier is not frozen to bedrock at the drilling site? This has implications on the thinning. Please clarify.

*Thanks for encouraging us to add a figure, since we were not sure whether the new results are important enough to add a figure or not. We now added a figure (Fig.5) reporting the entire annual layer counting, the most prominent identified time horizons, the $^{14}C$ data, and a ice flow fit based on the Nye law.*

*Our speculation on melting was meant in the sense to not exclude that this might had happened in the past, since Mikhalenko et al. (2015) proposed that in present time basal melting might occur below ice thicknesses of more than 220 m at the Elbrus site possibly due to a heat magma chamber. If ever this heat magma chamber had an increased energy in the past this underlying heat source could have melted the lowest ice layers, without influencing significantly above situated ice layers still having negative temperatures.*

*The manuscript was revised accordingly: "From the observed temperature gradient in the borehole ELB site, Mikhalenko et al. (2015) calculated a heat flux at the bottom glacier that is presently 4-5 times larger than the mean value for the Earth's surface, possibly due to a heat magma chamber of the Elbrus volcano, leading to potential basal ice melting when ice thicknesses exceeds 220 m. Though the ice at the drill site and upstream is at present frozen to bedrock, we can not exclude that in the past, assuming a more active heat chamber due to the eruption of 50 ± 50 CE (located 1.6 km away from the Eastern Elbrus plateau), a temporary basal ice melting had occurred at the drill site. If so, that may explain the young age of basal ice at the volcanic crater site compared to other non-volcanic mountain glaciers."*

I am not convinced by the equidistant binning of the summer and winter layers to obtain monthly values. This requires the absence of seasonality in precipitation and snow preservation. Precipitation data from nearest meteorological stations show strong seasonality (Kozachek et al., 2017). Since the monthly data are not really discussed, accept for showing the seasonality of chemical tracers in Fig. 6, I suggest deleting this part and the figure.

*We agree and the figure with monthly resolution was skipped. In addition we revised the text accordingly (see paragraph 2.2) and the figure caption of old figure 2 (now figure 3).*

*Figure 3. ELB ice chronology at depth intervals of 166.2 to 168.5 m (top), 154.4 to 156.5 m (mid), and from 99.8 to 107.3 m (bottom), based on the ammonium and succinate stratigraphy. Vertical red lines denote yearly dissection based on identification of winter layers (see Sect. 2.2). For the two oldest time-periods (1773-1782 and 1850-1859), each sample was 2 cm long whereas for the most recent time period (1925-1934) one sample was on average 4 cm long.*

*Note that, though being not coherent with the intra seasonal precipitation distribution (see Kozachek et al., 2017), we here assumed that the accumulation is equally distributed within summer and winter seasons. "*

Identification of annual layers and attribution of summer and winter layers: You use two criteria for that (ammonium and succinate). To which of the two do you give priority when the two signals do not agree? How does the attribution of summer and winter layers presented in this manuscript agree with the one based on the stable isotope record of the same core (Kozachek et al., 2017)?

*As now specified in the text, it is required that at least one of the criteria is fulfilled and the value of the other is below or close to the limit (< 10% above). "Requiring that at least one of the criteria is fulfilled and the value of the other is below or close to the limit (< 10% above) the annual counting was found to be very accurate dating (a 1-year uncertainty) over the last hundred years when anchored with the stratigraphy with the Katmai 1912 horizon (Mikhalenko et al., 2015)."*

*Kozachek et al. (2017) compared the dating derived from examination of the annual cycle of $^{18}O$ with the dating obtained with the ammonium-succinic criteria, and found a discrepancy of 2 years at a depth of 126 m (at the end of the examined water isotope profile). The text in paragraph 2.2 was changed as "A good agreement (a 2-year discrepancy) was also found when comparing this dating with the chronology achieved by annual layer counting of the water stable $^{18}O$ isotope back to 1900 (Kozachek et al., 2017). Though the annual counting becomes less evident prior to 1860, Mikhalenko et al. (2015) reported an ice age of 1825 at 156.6 m depth, what is still consistent with the presence of volcanic horizon at around 1833-1940 such as Coseguina (1835)."*

14C-dating: Was the AMS equipped with a gas ion source? You used a rather old version of Oxcal. I suggest using an updated version.

*Yes the AMS was equipped with a gas source. The Oxcal version used (4.3) is to our knowledge the actual one (see https://c14.arch.ox.ac.uk/oxcal.html), but the reference we used (Bronk Ramsey, 1995) refers to the overall introduction of Oxcal. In the revised version, we added the reference of Bronk Ramsey (2009), which deals with new features of Oxcal 4.*

*we reworded the sentences as "After cryogenic extraction of the CO2 content, radiocarbon analyses were done at the accelerator mass spectrometer facility at the Curt-Engelhorn-Center Archaeometry (CEZA) in Mannheim equipped with a Gas Interface System (GIS) (Hoffmann et al., 2017). Calibration of the retrieved 14C ages was done using OxCal version 4.3 (Bronk Ramsey, 1995, Bronk Ramsey, 2009)."*

Ion balance: Use the same unit (either ppb or uEq/L) in the text and in Figure 4.
*OK done*

Attribution of dust sources: This part of the manuscript is not convincing to me. What is the argument to relate high Ca concentrations to Saharan dust and low Ca concentrations to sources in the Middle East?

*The argument is given in the paper from Kutuzov et al. (2013) as well as in the companion paper as now referenced in the text.*

The plots in Figure 5 show a large scatter and low correlation coefficients, so I wonder if the ion ratios you discuss are significantly different. For the ions with strong anthropogenic influence this correlation analysis is anyway not meaningful without splitting the data set in

the pre-industrial and industrial periods. To me this part of the manuscript is weak, distracts from the main message, and could be omitted.

*We agree with you and the whole discussion on the origin of dust (including old Table 2 and old Figure 5) was removed.*

Important is to estimate the amount of sulfate originating from dust and correct for that when discussing anthropogenic sulfate. Attribution of sulfate related to mineral dust: Instead of arbitrarily introducing a Ca level to identify dust events, I propose to look at the pre-industrial Ca to sulfate correlation. If both are highly correlated, you can use this ratio to correct for mineral dust sulfate in the industrial period.

*Thanks for this very important comment. We agree and follow your suggestion. This part was totally reworked and new dust-free sulfate figures are shown.*

*"As discussed in section 3, large dust events significantly enhanced the sulfate level of the ELB ice. Since, as detailed in Kutuzov et al. (this issue), their occurrence have changed over the past with more frequent events after 1950, we have examined to what extent they contribute to the sulfate trend. It is difficult to accurately directly correct sulfate concentration from the large dust event contribution since the amount of sulfur trapped by the alkaline material during atmospheric transport towards the site would be very different from event to event. For instance, Kozak et al. (2012) reported non-sea-salt-sulfate to non-sea-salt calcium mass ratios in aerosol collected in the eastern Mediterranean ranging between 0.25 (in case of direct arrival at the site of air mass from the Sahara) to 1.15 (when mineral dust passed through polluted sites located in the Balkans and Turkey before arriving at the site). Instead of corrected sulfate values for the large dust events, we therefore have reported in Figure 8 individual values of total sulfate and sulfate calculated after having removed from the average samples suspected to contain large amount of dust ($SO_4^{2-}{}_{red.}$ values). The influence of large dust events on the long-term SSA winter trend is rather insignificant and if existing (i.e. effect of < 10 ppb) remaining limited to two decades around 1870 and the recent decade (2000-2010) (Fig. 9). Remaining negligible prior to 1850, the large dust event effect on the summer trend gradually increases after 1950, reaching often 100 ppb after 1960. This change of large dust events results from change in the occurrence of drought in North Africa and Middle East regions (Kutuzov et al., this issue).*

*In addition to the enhanced frequency of large dust events after 1950, the calcium background concentrations (i.e., $Ca^{2+}{}_{red}$) also change over time with an increase from 68 ± 21 ppb prior to 1900 CE to 194 ± 61 ppb after 1960 CE. As discussed by Kutuzov et al. (this issue), this change may result from changes in precipitation and soil moisture content in Levant region (Syria and Iraq). In view to discuss the free-dust sulfate changes with respect to anthropogenic emissions, we make an attempt to correct the sulfate record from this increase of the background level of dust. To do so, we examined in Figure 10 the relationship between calcium and sulfate concentration in individual summer samples corresponding to pre-industrial time (1774-1900 CE). Although being poor ($R^2$=0.32), the correlation suggests a mean slope of the linear SO₄red-Cared relationship close to 1. The use of this value to correct sulfate from background dust emissions would lead to an overestimation of the sulfate dust contribution. Indeed, as seen in Figure 10, there are numerous samples that contain more sulfate than what is expected with respect to the presence of pure calcium sulfate (gypsum, see the blue line reported in Figure 10), likely to due the presence of sulfate as ammonium sulfate or sulfuric acid. To correct sulfate from the background dust contribution we here have used a sulfate to calcium ratio close to 0.63 (see the red line drawn as the lower envelope of the relationship in Figure 10) and subtracted this contribution from the $SO_4^{2-}{}_{red.}$ values by using the $Ca^{2+}{}_{red.}$ values."*

I recommend adding a map with the Elbrus site, outlining the dust and SO2 emission source areas.

*OK, done: we show a map (Figure 1) where countries can be easily identified and dust emission are reported.*

Table 5 is mentioned in the text, but does not exist.

*Sorry (it was a typographic error: Table 3)*

Discussion of outliers: This is hard to follow without seeing the raw data (which should be shown anyway). Can some of the outliers be explained by volcanic events? It is strange that you don't see a signal of the largest eruption in the last centuries (Tambora, 1815) and the largest eruption in the Northern Hemisphere in the last centuries (Laki, 1783).

*In the revised version we introduce a raw data figure (now Fig. 7). Well we were also surprised by the lack of evidence for Laki and Tambora.*

Comparison with emission estimates: You stress in the manuscript the importance to distinguish between summer and winter sulfate values and trends (to me the trends look similar). And then you compare this with emission estimates, which are annual values (I guess). This is inconsistent. You need to include the total anthropogenic sulfate record, which would also be very valuable for comparison with data sets from other ice cores, which are not resolved in summer and winter values. In addition, you give the impression that SO2 emissions in winter are much lower than in summer. The opposite is the case. The major factor producing the difference in summer and winter values at high-alpine sites is the reduced vertical atmospheric transport in winter (and not the variation in source area). You need to explain this in the manuscript.

*Indeed the winter and summer trends appears quite similar at ELB. This contrasts to the case of CDD for which the model EMEP simulations confirmed the finding of a clear difference between summer and winter. We cannot discuss further that since we don't have EMEP simulations for the ELB ice core. Anyway in the revised version we also report annual means and discuss as follows: "Since available at all three sites, we compared in Figure 11 the annual long-term trends of dust-free sulfate from ELB, CDD, and BEL. The ELB and CDD annual values were calculated as arithmetic mean from 5 yr-SSA winter and summer records, whereas the BEL annual data refer to 5 year averaged raw data. Examination of the three annual records reveal three major differences between the three sites: (1) an impact of anthropogenic emissions already significant in 1910-1930 at CDD but neither at ELB nor at BEL, (2) a maximum of the anthropogenic perturbation which is reached in 1970-1980 at CDD and later at the two other sites (10 years after at ELB, and a few years after at BEL), and (3) a far less pronounced re-decrease at the beginning of the 21$^{th}$ century at ELB compared to CDD. The re-decrease of sulfate over the very recent decades is somewhat stronger at BEL than at ELB. Using data from Smith et al. (2011), available at http://sedac.ciesin.columbia.edu/data/set/haso2-anthro-sulfur-dioxide-emissions-1850-2005-v2-86, we report in Figure 12 emissions of SO$_2$ from countries located nearby the ELB site: Georgia, Azerbaijan, Syria, Irak, Turkey, Russian, Iran or located further north (Ukraine) and west (Bulgaria). In these countries SO$_2$ emissions became significant after 1930 and reached maximum in the late 80's or later (for Turkey and Iran). This feature clearly differs from the situation at CDD where emissions from countries located around the site (France, Italy, Spain, Switzerland and Germany) were already significant in 1930 and exhibited a maximum between the early 70's and the early 80's (Figure 10). For BEL, Eichler et al. (2009) demonstrated the importance of emissions from eastern Europe for the dust-free sulfate annual record at that site.*

*Finally, we tentatively examine the cause of the recent decrease of sulfate, focusing on the summer season for which the most relevant source regions are limited to countries located nearby the site. As discussed by Kutuzov et al. (this issue), 10 day backward air mass trajectories calculated for the ELB site using the NOAA HYSPLIT-4 model suggest that, in summer, air masses arriving at ELB mainly originate from the nearby Georgia, Azerbaijan, Syria, Irak, and from Turkey, South Russian, and North of Iran. As previously discussed by Fagerli et al. (2007), the CDD site in summer is mainly influenced by emissions from France, western Germany, Italy and Spain. Consistently with $SO_2$ emission changes, the recent sulfate decrease is more pronounced at CDD than ELB with a recovered 2005 level (316 ppb) close to the 1950 one (296 ppb) (Figure 13). At the ELB site this is not the case, here the 2005 level (380 ppb in 2005) is found to be still almost two times higher than the one of 1950 (227 ppb in 1950). An intermediate pattern is seen at BEL, likely due to a weak impact of countries like Turkey that significantly contribute to the ELB record but not the BEL one, and a more strong contribution of emissions from Russian at BEL (see also Eichler et al., 2009) than at ELB."*

Considering the SO2 emission source areas you identified it is strange that you just compare the Elbrus record with the CDD record from the Alps. I strongly recommend to include the sulphate record from Eastern Europe (from Belukha ice core, Eichler et al., EST 2012). *Thanks to rise this important question. We agree and now compare ELB, CDD, and Belukha records (see our answer above and the new Figure 11).*

Figure 1. I don't see the point of showing the mean summer and winter sample length. This should not be so different from the sample resolution.
*OK this aspect is very important for the discussion made in the companion paper of the enhanced frequency of sporadic large dust events after 1950. So we remove this to the companion paper and reworded the old figure 1 (showing now annual, half-year summer and winter ice thickness).*

Technical corrections
Title: seems too long and a bit cumbersome. Suggestion: Reconstruction of anthropogenic sulfate trends from Elbrus ice core, Caucasus.
*OK we changed it: The Elbrus (Caucasus, Russia) sulfate ice core record: reconstruction of past anthropogenic sulfur emissions in south-eastern Europe".*

Abstract L. 18: After having examined. . . Rephrase and give the results: dust contribution to sulfate concentrations was identified and subtracted to focus on anthropogenic sulphate (not sulfur).
*OK, this sentence has been reworded.*

P2L4: Replace Andreae et al., 2015 with a newer estimate e.g. from IPCC.
*Well we prefer to cite this reference, which highlighted for the first time this aspect instead of an updated report (we don't discuss further this aspect later in the paper).*
*P2L8: "impact" instead of "disturb". Ok, we changed to "impact"*

P2L13-15: The Altai and Kamchatka are not part of Europe.
*We correct: In Eurasia…"*

P2L16-19: ice cores have been investigated . . .to examine.
*Ok done.*

P3L12: Give more details how ice cores were decontaminated (by removing xx cm from the outside of the core. . .)
*Ok the section was revised as following: "Ice cores were subsampled and decontaminated at -15°C using the electric plane tool methodology described in Preunkert and Legrand (2013). In brief, in a first step ice samples were cut with a band saw. After that, all surfaces of the cut samples were decontaminated by removing ~ 3 mm with a pre-cleaned electric plane tool under a clean air bench."*

P3L17:loss
*Ok done*

P3L29: Give details, which fluid was used.
*Ok we added the name of the drilling fluid: "During the drill operations, an incident occurred at the depth of 31 m and a fluid (Havoline XLC, Texaco company) was poured in the hole to liberate the drill device."*

P4L7: For the ammonium seasonality earlier work should be cited (Maupetit et al., Atmos. Environ., 1995; Eichler et al., JGlac., 2000):
*OK done*

P6L14-15: Replace "disturb the chemistry" by changes the chemical composition Table 1: Include 14C lab sample reference number.
*Ok disturb was replaced, and in fact the "Sample name" we gave in Column 1 of Table 1 is the 14Csample name used in the 14C lab. The column was renamed.*

**The Elbrus (Caucasus, Russia) sulfate ice core record: reconstruction of past anthropogenic sulfur emissions in south-eastern Europe**

[revised manuscript text omitted]

of the industrialization.

Here we report on the glaciochemistry of a deep ice core  drilled to bedrock at 182.6 m (142.1 mwe) in 2009  at Mount Elbrus  in the Caucasus, Russia. ~~Glaciological settings of the drill site are detailed in (Mikhalenko et al., 2015). In Brief, the spatial size of the glacier plateau is about 0,5km², and the surface snow accumulation at the drill site is about 1,5 mwe yr⁻¹. Ice penetrating radar measurements made in 2007 and 2009 revealed a maximum glacier thickness of 255 ± 8 m at the central part of the plateau, and minimum values of about 60 m near the western border of the glacier. Borehole temperature measurements ranged from −17 °C at 10 m depth to −2.4 °C at 181.8 m. Occasionally melting of surface snow can occur, however, the thickness of the infiltration ice layers, which do not form every year, does not exceed 10 mm.~~

Seasonally resolved chemical records were obtained back to 1774 (i.e., well prior to the onset of the industrial period). Data are discussed in two companion papers of which this one. The present paper examines first of all the impact of _large_ dust plumes, which arrive sporadically from Sahara and Middle East, on the chemical composition of the Elbrus (ELB) snow and ice layers. It then focuses on long-term _dust-free_ sulfate trends in relation to growing sulfur pollution. The long-term summer and winter trends of _dust-free_ sulfate are discussed with respect to past $SO_2$ emissions in _south-eastern_  Europe and compared to those extracted at the Alpine site of CDD  and the Siberian Altai (Belukha glacier) in relation to $SO_2$ emissions from western and eastern Europe, respectively. The second paper focuses on calcium (a dust tracer) long-term trend (Kutuzov et al., this issue), discussing its past changes in relation with natural variability, as well as climatic and land use changes in the dust source regions Middle East and North Africa.

**2 Methods and Dating**

A deep ice core was drilled to bedrock (182.6 m, i.e. 142.1 meter water equivalent (mwe)) in 2009 on the western plateau of Mt. Elbrus (43°21′N, 42°26′E; 5115 m above sea level) in the Caucasus, Russia (Fig. 1). Glaciological settings of the drill site are detailed in Mikhalenko et al. (2015). In brief, the surface of the glacier plateau is about 0.5 km², and the surface snow accumulation at the drill site is about 1.5 mwe yr⁻¹. Ice-penetrating radar measurements made in 2007 and 2009 revealed a maximum glacier thickness of 255 ± 8 m at the central part of the plateau, and minimum values of ~60 m near the western border of the glacier. Borehole measurements indicated temperatures of -17°C at 10 m depth and -2.4°C at 181.8 m depth. Occasionally melting of surface snow can occur, however, the thickness of the infiltration ice layers, which do not form every year, does not exceed 10 mm. After the overall presentation from Mikhalenko et al. (2015), two other studies of the

ELB ice core were dedicated to black carbon (Lim et al., 2017) and water stable isotope composition on the 126 m upper layers (Kozachek et al., 2017).

**2.1 Discrete Subsampling of firn and ice, and Chemical Analysis**

Ice cores were subsampled and decontaminated at -15°C using the electric plane tool methodology described in Preunkert and Legrand (2013). In brief, in a first step ice samples were cut with a band saw. After that, all surfaces of the cut samples were decontaminated by removing ~ 3 mm with a pre-cleaned electric plane tool under a clean air bench.  A total of 3724 subsamples were obtained along the upper 168.6  m (131.6 m-we) of the Elbrus core.  As expected from  glacier ice flow (e.g.  Paterson and Waddington, 1984), a decrease of  the  annual ice thickness with depth is observed ( Fig. 2). Annual layer thicknesses decreases  from 1.5 mwe (0.8 mwe in summer and 0.7 mwe in winter) near the surface, to 0.46 mwe at ~100 m (i.e. 75 mwe) (0.35 mwe in summer and 0.11 mwe in winter), 0.21 mwe at 155 m (i.e. 120 mwe) (0.12 mwe in summer and 0.09 mwe in winter) and 7we (i.e. 122 mwe) depth.  To minimize the loss of temporal resolution with depth along the core, the sample depth resolution was decreased from 10 cm at the top to 5 cm at 70 m (47 m we) and 2 cm at 157 m (122 mwe) depth and below.  In this way, an average of 9 summer samples per year were sampled at 157 m depth (compared to 15 summer samples per year near the surface). The large decrease of the net snow accumulation in winter below 155 m (Fig. 2), likely due to more wind erosion of winter than summer snow layers upstream the drill site as already observed at other high altitude glacier sites (e.g. Preunkert et al., 2000),  leads to a more pronounced loss of resolution in these winter layers compared to the surface layers (12 samples per winter near the surface and 1-2 samples per winter at 157 m depth).

[revised manuscript text omitted]

15  at the bottom glacier that is presently 4-5 times larger than the mean value for the Earth's surface, possibly due to a heat magma chamber of the Elbrus volcano,  leading to potential basal ice melting when ice  thicknesses exceeds  220 m.

Though the ice at the drill site and upstream  is at present frozen to bedrock, we can  not exclude that in the past, assuming a more active heat chamber due to the eruption of 50 ± 50 CE (located 1.6 km

20  away from the Eastern Elbrus plateau), a temporary basal ice alsoThat may have letto a melting and removing of the basal without influencing the above situated ice layers and If so, that may explain the young age of basal ice at the volcanic crater site compared to other non-volcanic mountain glaciers. The age of the basal ELB ice is nevertheless largely greater than expected by ice flow model calculations, estimating a basal ice age of

25  less than 400 years at the drill site (Mikhalenko et al. 2015). Fig. 5 summarizes  the extended depth-age relation, including annual layer counting back to 1774 CE, the prominent time horizons, and PO$^{14}$C data. To interpolate data to a continuous age-depth relation, a two-parameter fit (based on a simple analytical expression for the decrease of the annual layer thickness with depth) was used (Nye, 1963; Jenk et al., 2009; Preunkert 
[revised manuscript text omitted]
. In conclusion, large dust events significantly influence the chemistry of ELB ice for many species, requiring a case by case examination depending on the nature of the dust contribution: primary emissions for sodium and other cations (except ammonium), neutralization of the alkaline material during atmospheric transport for nitrate, and both primary gypsum emissions and neutralization of the alkaline material by acidic sulfur during atmospheric transport for sulfate, for instance.

**4 Long-term summer and winter trends of sulfate in the Elbrus ice**

From the winter/summer dissection made on the basis of the ammonium and succinate stratigraphy (Sect. 2), monthly means as well as half year summer and winter means were calculated over the 1774 to 2010 period. In Figure 6, we report the seasonal cycle of sulfate, ammonium, and succinate averaged across a pre-industrial period (1775-1825 AD) and two different periods of the industrial period (1940-1960 and 1980-2000 AD). Individual sulfate half year summer and winter means are reported in Figure 7, considering raw sulfate data and those regarded as free of dust ($SO_{4\ red.}^{2-}$), respectively.

A few outliers of unknown origin were observed in the sulfate raw data set including 22 ppm at 166.65 m depth (summer 1780 AD), 2248 ppb at 146.38 m depth (summer 1862 AD), 1080 ppb at 146.11 m depth (summer 1863 AD), and 864 ppb at 154.73 m depth (winter 1833/34 AD) (Fig. 7). These individual values were removed when calculating the corresponding half-year summer and winter means reported in Fig.ure 78. In addition, single winter samples with sulfate levels of 815 ppb at 160.62 m depth (winter 1810/11 AD) and 3 data points from 150 to 230 ppb corresponding to winters 1786-87, 1827-28, and 1844-45 were not considered and corresponding half-year winter values were not reported.

A few ELB snow and ice layers are impacted by known volcanic eruptions. As discussed by Mikhalenko et al. (2015), ice layers dated to 1911 and 1913 were probably impacted by the 1912 AD Katmai eruption, and summer layers of 1836 and 1837 by the 1835 AD Coseguina eruption, respectively. In addition, we suspect the 1854 AD Shiveluch eruption to have impacted summer 1854 ice layer and finally, although less evident since this part of the core is made up of splitted ice (see

ect 2), the Cotopaxi 1877 AD eruption may have influenced the winter 1877/1878 layer. To discuss the long-term trends of sulfate in relation to growing $SO_2$ emissions, these half-year summer and winters means suspected to be contain volcanic debris, were discarded in Fig 9. To minimize the effect of year-to-year variability due to meteorological transport conditions we added the first component of single spectra analysis (SSA) with a five-year time window in Fig

5  9.

As discussed in Sect. 3, large dust events significantly enhanced the sulfate level of the ELB ice. Since, as detailed in Kutuzov et al. (this issue), their occurrence have changed over the past with more frequent events after 1950, we have examined to what extent they contribute to the sulfate trend. It is difficult to accurately directly correct sulfate concentration from the large dust event contribution since the amount of sulfur trapped by the alkaline material during atmospheric

10  transport towards the site would be very different from event to event. For instance, Kozak et al. (2012) reported non-sea-salt-sulfate to non-sea-salt calcium mass ratios in aerosol collected in the eastern Mediterranean ranging between 0.25 (in case of direct arrival at the site of air mass from the Sahara) to 1.15 (when mineral dust passed through polluted sites located in the Balkans and Turkey before arriving at the site). Instead of corrected sulfate values for the large dust events, we therefore have reported in Fig. 8 individual values of total sulfate and sulfate calculated after having removed from the

15  average samples suspected to contain large amount of dust (SO$_4^{2-}$red. values). The  influence of large dust events on the long-term SSA winter trend is rather insignificant and if existing (i.e. effect of < 10 ppb) remaining limited to two decades around 1870 and the recent decade (2000-2010) (Fig. 9). Remaining negligible prior to 1850, the large dust event effect on the summer trend gradually increases after 1950, reaching often 100 ppb after 1960. This change of large dust events results from change in the occurrence of drought in North Africa and Middle East regions (Kutuzov et al., this issue).

20   In addition to the enhanced frequency of large dust events after 1950, the calcium background concentrations (i.e., Ca$^{2+}$red.) also changed over time with an increase from 68 ± 21 ppb prior to 1900 CE to 194 ± 61 ppb after 1960 CE. As discussed by Kutuzov et al. (this issue),

25  this change may result from changes in precipitation and soil moisture content in Levant region (Syria and Iraq). In view to discuss the free-dust sulfate changes with respect to anthropogenic emissions, we make an attempt to correct the sulfate record from this increase of the background level of dust. To do so, we examined in Fig. 10 the relationship between calcium and sulfate concentration in individual summer samples corresponding to pre-industrial time (1774-1900 CE). Although being poor ($R^2$=0.32), the correlation suggests a mean slope of the linear SO$_4$red-Cared relationship close to 1. The use of

30  this value to correct sulfate from background dust emissions would lead to an overestimation of the sulfate dust contribution. Indeed, as seen in Fig. 10, there are numerous samples that contain more sulfate than what is expected with respect to the presence of pure calcium sulfate (gypsum, see the blue line reported in Fig. 10), likely to due the presence of sulfate as ammonium sulfate or sulfuric acid. To correct sulfate from the background dust contribution we here have used a sulfate to calcium ratio close to 0.63 (see the red line drawn as the lower envelope of the relationship in Fig. 10) and subtracted this

contribution from the $SO_4^{2-}_{red.}$ values by using the $Ca^{2+}_{red.}$ values.

As seen in Fig. 89, the derived mean summer and winter pre-industrial free-dust sulfate levels in ELB ice (taken as the mean value observed from 1774 to 1850) is of  70 ppb and  40 ppb, respectively . In summer as in winter, the free-dust sulfate  values remained close to the pre-industrial values until 1910-1920 ( 81 ppb instead of  70 ppb in summer,  47 ppb in winter). After 1920, dust-free sulfate levels increased at a mean rate of  5 ppb per year  in summer, and  1. ppb yr$^{-1}$  in winter. The sulfate increase then accelerated between 1950 and 1975 ( 10 ppb yr$^{-1}$ in summer, 5 ppb yr$^{-1}$ in winter), until a maximum of  530 ppb in summer ( 255 ppb  in winter) was reached at the end of the 80's. After 1990, sulfate levels decreased to  390 ppb in summer ( 154 ppb  in winter) during the first decade of the twenty first century.

**5 Comparison between Elbrus Alpine, and Siberian Altai long-term sulfate trends**

**5.1 The Alpine CDD and Siberian Altai (Belukha, BEL) ice core Sulfate Records**

The ELB dust-free sulfate long-term trend is compared with those previously extracted from the Alpine CDD site (ice cores denoted C10 and CDK in Fig. 911). C10 sulfate data were presented in Preunkert et al. (2001), and those form CDK in (Legrand et al., 2013). Since winter data from CDD are more limited (only a few pure winter layers are available between 1890 and 1930, Legrand et al., (2018) we here focus on the comparison of summer levels. The two CDD cores were dated by annual layer counting using the pronounced seasonal variations of ammonium. The two chronologies were in excellent agreement over their overlapping period from 1925-1990 (Legrand et al., 2013; Preunkert et al., 2000). A re-evaluation of the C10 chronology based on very recently made continuous measurements of heavy metals, as well as a comparison to a well-dated Greenland ice core record (McConnell and Edwards, 2008), resulted in a revised C10 chronology (Legrand et al., 2018). As for C10, continuous measurements of heavy metals are also available in the lowest part of CDK (Preunkert et al., 2019). It was thus possible to identify the distinct Greenland increases of thallium, lead, and cadmium associated with the widespread start of coal burning at the beginning of the Industrial Revolution in 1890 CE also in the CDK core (at 117.8 m (90.5 mwe)). This time marker was then used to constrain a revised annual layer counting in the early 20$^{th}$-century part of the CDK record.  The dust-free ELB sulfate long-term trend is also compared with the one previously extracted from the Siberian Altai (Belukha glacier, denoted BEL in Fig. 11) by Eichler et al. (2009 and 2012).

**5.2 ELB versus CDD and BEL sulfate trends**

Since available at all three sites, we compared in Fig. 11 the annual long-term trends of dust-free sulfate from ELB, CDD, and BEL. The ELB and CDD annual values were calculated as arithmetic mean from 5 yr-SSA winter and summer records, whereas the BEL annual data refer to 5 year averaged raw data. Examination of the three annual records reveal three major differences between the three sites: (1) an impact of anthropogenic emissions already significant in 1910-1930 at CDD but neither at ELB nor at BEL, (2) a maximum of the anthropogenic perturbation which is reached in 1970-1980 at CDD and later at the two other sites (10 years after at ELB, and a few years after at BEL), and (3) a far less pronounced re-decrease at the beginning of the 21$^{th}$ century at ELB compared to CDD. The re-decrease of sulfate over the very recent decades is somewhat stronger at BEL than at ELB.

Using data from Smith et al. (2011), available at http://sedac.ciesin.columbia.edu/data/set/haso2-anthro-sulfur-dioxide-emissions-1850-2005-v2-86, we report in Fig. 12 emissions of $SO_2$ from countries located nearby the ELB site: Georgia, Azerbaijan, Syria, Irak, Turkey, Russian, Iran or located further north (Ukraine) and west (Bulgaria). In these countries $SO_2$ emissions became significant after 1930 and reached maximum in the late 80's or later (for Turkey and Iran). This feature clearly differs from the situation at CDD where emissions from countries located around the site (France, Italy, Spain, Switzerland and Germany) were already significant in 1930 and exhibited a maximum between the early 70's and the early 80's (Fig. 10). For BEL, Eichler et al. (2009) demonstrated the importance of emissions from eastern Europe for the dust-free sulfate annual record at that site.

~~For summer (see Figure 9 a and b), the pre-industrial sulfate ELB value ($SO_4^{2-}$red. = 113 ppb) thus exceeded the CDD one (66 ppb) (Preunkert et al., 2000). A similar difference is observed for winter with $SO_4^{2-}$red. close to 68 ppb at ELB (Figure 7) compared to 20 ppb observed by Preunkert et al. (2000) at CDD. It is out of the scope of this work to discuss the cause of this difference between the two ice cores but we can first mention the existence of local volcanic sulfur emissions (as evidenced by direct on site observations of a sulfur smell nearby the ELB drill site). The pre-industrial summer level of dust free calcium samples at ELB (74 ppb, Kutuzov et al., this issue) is higher than the one at CDD (45 ppb, Legrand, 2002). That may also contribute to the ELB/CDD difference of the sulfate pre-industrial level. Clearly, more work, including simulations with transport and chemistry models considering also oceanic emissions of DMS may help here.~~

~~Figure 9 compares the increasing summer sulfate trends of the ELB and CDD sites. Three major differences between the two sites are revealed: (1) an impact of anthropogenic emissions already significant in 1910 at CDD and not at ELB, (2) a maximum of the anthropogenic perturbation from 1970 to 1980 at CDD and 10 years after (1980-1990) at ELB, and (3) a far less pronounced re-decrease at the beginning of the 21$^{th}$ century at ELB compared to CDD.~~ Finally, we tentatively examine the cause of the recent decrease of sulfate, focusing on the summer season for which the most relevant source regions are limited to countries located nearby the site.

As discussed by Kutuzov et al. (this issue), 10 day backward air mass trajectories calculated for the ELB site using the NOAA HYSPLIT-4 model suggest that, in summer, air masses arriving at ELB mainly originate from the nearby Georgia,

Azerbaijan, Syria, Irak, and from Turkey, South Russian, and North of Iran. As previously discussed by Fagerli et al. (2007), the CDD site in summer is mainly influenced by emissions from France, western Germany, Italy and Spain. Consistently with SO$_2$ emission changes, the recent sulfate decrease is more pronounced at CDD than ELB with a recovered 2005 level (316 ppb) close to the 1950 one (296 ppb) (Fig. 13). At the ELB site this is not the case, here the 2005 level (380 ppb in 2005) is found to be still almost two times higher than the one of 1950 (227 ppb in 1950). An intermediate pattern is seen at BEL, likely due to a weak impact of countries like Turkey that significantly contribute to the ELB record but not the BEL one, and a more strong contribution of emissions from Russian at BEL (see also Eichler et al., 2009) than at ELB.

We report in Figure 10 emissions of SO$_2$ from these countries and from a few others located further north (Ukraine) and west (Bulgaria). In these countries SO$_2$ emissions reached maximum in the late 80's or later (for Turkey and Iran). This feature clearly differs from the situation at CDD where countries around the site (France, Italy, Spain, Switzerland and Germany), thought to be the main contributors for sulfate in CDD ice (Fagerli et al., 2007), exhibit a maximum between the early 70's and the early 80's (Figure 10).

On this basis and as a first attempt, we compare the ELB and CDD summer sulfate trends with SO$_2$ emissions from surrounding countries. It can be seen that the impact of growing anthropogenic SO$_2$ emissions started later at ELB (after 1920) compared to CDD (after 1900). The 10-year delay of the sulfate maximum at ELB compared to CDD is also well seen in the enhancement course of SO$_2$ emissions. Note also that as indicated by the emissions, the maximum enhancement at ELB (550 ppb between 1980 and 1990) is slightly weaker that the one at CDD (665 ppb between 1974 and 1984) (Table 3). Finally, consistently with SO$_2$ emission changes, the recent sulfate decrease is more pronounced at CDD than ELB with a recovered 2005 level (254 ppb) close to the 1950 one (234 ppb). At the ELB site this is not the case, here the 2005 level (380 ppb in 2005) is found to be still around two times higher than the one of 1950 (180 ppb in 1950).

**6. Conclusions**

Based on the ammonium and succinate stratigraphy, the upper 168.6 m of the deep ice core extracted at Mt Elbrus (Caucasus) in 2009 were dated by counting annual layers back to 1774 CE. The derived seasonally resolved chemical records cover the years 1774-2009 making this ice core particularly useful to reconstruct many aspects of atmospheric pollution in south-eastern central Europe from pre-industrial times (1850 CE) to present-day. Below 169 m depth the annual counting is not possible but radiocarbon analysis of the particulate organic carbon fraction in the basal ice of the glacier suggests an age of ~1670 ± 400 cal yr BP. We have examined the impact on the chemical composition of the Elbrus ice layers of arrival at the site of large dust plumes originating from Sahara and Middle East. We then report on the dust-free sulfate records sulfur pollution. The ELB dust-free sulfate record indicates a four six- and six seven-fold increase from prior to 1900 to 1980-1995 in winter and summer, respectively. Still moderate at the beginning of the 20$^{th}$ century, the sulfate increase accelerated after 1950, dust-free annual levels reaching a maximum in 1980-1990 (376 ± 10 ppb 730 ± 152 ppb in summer) and subsequently decreasing to 270 ± 18 ppb 630 ± 130 ppb in summer at the beginning of the 21$^{th}$ century. These

long-term sulfate changes observed in the ELB ice cores are compared with those previously obtained in Alpine and Siberian Altai ice. Consistently with past $SO_2$ emission inventories, a much earlier onset and a more pronounced decrease of the sulfur pollution over the three last decades is are observed in western than south-eastern and eastern central Europe.

**Data availability**

Sulfate and calcium data can be made available for scientific purposes upon request to the authors (contact: suzanne.Preunkert@univ-grenoble-alpes.fr or michel.legrand@univ-grenoble-alpes).

**Author contributions**

S. Preunkert and M. Legrand performed research, analyzed ice samples and data, and wrote the original manuscript. S. Kutuzov performed research, analyzed data, and commented original manuscript. P. Ginot and V. Mikhalenko performed analysis and commented original manuscript. R. Friedrich analyzed ice samples and commented original manuscript.

**Acknowledgments.**

The study was supported by the RSF grant 17-17-01270. The « Les Enveloppes Fluides et l'Environnement- Chimie Atmosphérique » (CNRS) program entitled "Evolution séculaire de la charge et composition de l'aérosol organique au dessus de l'Europe" (1262c008 –ESCCARGO) provided funding for ion chromatography analysis, with the support of Agence de l'Environnement et de la Maîtrise de l'Energie, and the LIA Vostok provided funding for $PO^{14}C$ measurements, respectively. Thanks a lot to Simon Escalle for preparing the blank ice for the $PO^{14}C$ measurements. We thank M. Schwikowski and the anonymous reviewer for very useful comments, which significantly improved the quality of the manuscript.

[Figure]

**Figure** 2 **(a) and (b):** Mean summer (a) and winter (b) half-**year** ice **thickness** along the Elbrus deep ice core. (c) Mean **annual** ice **thickness** and (d) summer to winter ratio **of** ice **th**ickness.

[Figure]

Figure 23. ELB ice chronology at depth intervals of 166.2 to 168.5 m (top), 154.4 to 156.5 m (mid), and from 99.8 to 107.3 m (bottom), based on the ammonium and succinate stratigraphy. Vertical red lines denote yearly dissection based on identification of winter layers (see Sect.Sect. 2.2). For the two oldest time-periods (1773-1782 and 1850-1859), each sample was 2 cm long whereas for the most recent time period (1925-1934) one sample was on average 4 cm long. Note that, though being not coherent with the intra seasonal precipitation distribution (see Kozachek et al., 2017), we here assumed that the accumulation is equally distributed within summer and winter seasons.

[Figure]

**Figure 34.** Distribution of succinate and ammonium concentrations observed in the deepest ice layers for which annual counting was possible (i.e., above 168.5 m depth, top) and 10 m below (bottom). The vertical red dashed bars denote the values of the winter criteria.

[Figure]

**Figure 5.** Depth (in mwe) age-relation of the ELB ice core derived from annual layer counting, prominent time horizons, and mean blank corrected and calibrated PO$^{14}$C data with 1σ age ranges. To interpolate data to a continuous age-depth relation a two-parameter fit was used following Nye (1963).

[Figure]

**Figure 46.** Mean ionic content of ELB layers deposited between 1950 and 1980. A and B: Mean composition of dust event samples containing more and less than 600 ppb (i.e., 30 µEq L$^{-1}$) of calcium. B and C: Mean composition of dust event samples containing less than 600 ppb of calcium compared to samples free of dust. Abbreviations Monoac. and Diac. stand for $C_1$-$C_3$ monocarboxylates and $C_2$-$C_5$ dicarboxylates, respectively (see Eqs. 2 and 3 in Sect. 3).

[Figure]

**Figure 7**. Raw sulfate data. The red dots denote the outliers, which were removed prior to calculation of the half-year summer and winter means. We also indicate samples possibly impacted by the volcanic eruptions of Cotopaxi and Shiveluch (see text).

[Figure]

5    **Figure 8.** Individual summer (red) and winter (blue) half-year means of sulfate along the ELB ice core. Solid circles refer to values calculated considering all samples. Open circle data (SO4 red.) were calculated after having removed samples considered to be impacted by  dust events (see ect. 3).

[Figure]

5 **Figure 89.** ELB half-year  summer and winter sulfate  trends from 1774 to 2010 (the first SSA component with a five-year time window, see text). The black lines refer to raw sulfate values, the dashed lines to sulfate means  calculated after removal of dust samples (SO4red.), the solid red and blue lines to sulfate values after having corrected SO4red values from the background dust contribution (see Sect. 3).
10  Samples suspected to be impacted by volcanoes were removed.

[Figure]

**Figure 10.** Sulfate versus calcium concentrations in summer samples free of large dust events (Red. values) in ice deposited prior to 1900. The blue line refers to a pure gypsum composition. The red line illustrates a lower envelope of the sulfate to calcium relationship.

[Figure]

**Figure 11**. Dust-free sulfate (annual values in black, summer in red, and winter in blue) trends at the CDD (a), ELB (b), and BEL (c) sites. For CDD we report the records derived from the C10 (dashed lines) and CDK (solid lines) ice cores (see Sect. 5.1). Green areas indicate the decades in which the sulfate levels reached their maxima.

**Figure 9.** Comparison of the dust corrected long-term sulfate summer trends ($SO_4^{2-}_{red}$ values) from two CDD ice cores (C10 and CDK) (a) and the ELB ice core (b) with $SO_2$ emissions from countries suspected to contribute to sulfate depositions at the two sites (c for CDD, d for ELB, see discussions in Section 5). (a) and (b): solid lines refer to the first SSA component with a five-year time window. Dashed blue lines refer to the respective pre-industrial sulfate levels (see section 5). (c): $SO_2$ emissions from France, Italy, Spain, Switzerland and half from Germany. (d): $SO_2$ emissions from Turkey, Georgia and Azerbaijan, half from Russia, a quarter from Ukraine, Bulgaria, and Iran. Green areas indicate the decades in which the sulfate levels reached their maxima.

[Figure]

[Figure]

**Figure 1012.** SO$_2$ emissions (from 1850 to 2005) from various countries located around the Caucasus (top) and the Alps (bottom). GEO&AZE&IRK&SYR denotes emissions from Georgia, Azerbaijan, Irak, and Syria. Data are from Smith et al. (2011).

[Figure]

**Figure 13.** Comparison of the dust-free sulfate summer trends (SO42-red. values) from two CDD ice cores (C10 and CDK) (a) and the ELB ice core (b) with SO2 emissions from countries suspected to contribute to sulfate depositions at the two sites (c for CDD, d for ELB, see discussions in Sect. 5.2). (a) and (b): solid lines refer to the first SSA component with a five-year time window. Horizontal dashed blue lines refer to the respective pre-industrial sulfate levels. (c): SO2 emissions from France, Italy, Spain, Switzerland and half from Germany. (d): SO2 emissions from Turkey, Georgia and Azerbaijan, half from Russia, a quarter from Ukraine, Bulgaria, and Iran. Green areas indicate the decades in which the sulfate levels reached their maxima.